# Microbial community composition and abundance after millennia of submarine permafrost warming

Julia Mitzscherling[1], Fabian Horn[1], Maria Winterfeld[2], Linda Mahler[1], Jens Kallmeyer[1], Pier P. Overduin[3], Lutz Schirrmeister[3], Matthias Winkel[4], Mikhail N. Grigoriev[5], Dirk Wagner[1,6] and Susanne Liebner[1,7]

[1]GFZ German Research Centre for Geosciences, Helmholtz Centre Potsdam, Section 3.7 Geomicrobiology, 14473 Potsdam, Germany
[2]Alfred Wegener Institute, Helmholtz Centre for Polar and Marine Research, Marine Geochemistry, 27570 Bremerhaven, Germany
[3]Alfred Wegener Institute, Helmholtz Centre for Polar and Marine Research, Permafrost Research, 14473 Potsdam, Germany
[4]GFZ German Research Centre for Geosciences, Helmholtz Centre Potsdam, Section 3.5 Interface Geochemistry, 14473 Potsdam, Germany
[5]Siberian Branch, Russian Academy of Sciences, Mel'nikov Permafrost Institute, Yakutsk, Russia
[6]University of Potsdam, Institute of Geosciences, 14476 Potsdam, Germany
[7]University of Potsdam, Institute of Biochemistry and Biology, 14476 Potsdam, Germany

*Correspondence to*: Susanne Liebner (Susanne.Liebner@gfz-potsdam.de)

**Abstract.** Warming of the Arctic led to an increase of permafrost temperatures by about 0.3°C during the last decade. Permafrost warming is associated with increasing sediment water content, permeability and diffusivity and could on the long-term alter microbial community composition and abundance even before permafrost thaws. We studied the long-term effect (up to 2500 years) of submarine permafrost warming on microbial communities along an onshore-offshore transect on the Siberian Arctic Shelf displaying a natural temperature gradient of more than 10 °C. We analysed the in-situ development of bacterial abundance and community composition through total cell counts (TCC), quantitative PCR of bacterial gene abundance and amplicon sequencing, and correlated the microbial community data with temperature, pore water chemistry and sediment physicochemical parameters. On time-scales of centuries, permafrost warming coincided with an overall decreasing microbial abundance whereas millennia after warming microbial abundance was similar to cold onshore permafrost. In addition, the dissolved organic carbon content of all cores was lowest in submarine permafrost after millennia-scale warming. Based on correlation analysis TCC unlike bacterial gene abundance showed a significant rank-based negative correlation with increasing temperature while bacterial gene copy numbers showed a strong negative correlation with salinity. Bacterial community composition correlated only weakly with temperature but strongly with the porewater stable isotopes δ18O and δD, and with depth. The bacterial community showed substantial spatial variation and an overall dominance of Actinobacteria, Chloroflexi, Firmicutes, Gemmatimonadetes and Proteobacteria which are amongst the

microbial taxa that were also found to be active in other frozen permafrost environments. We suggest that, millennia after permafrost warming by over 10°C, microbial community composition and abundance show some indications for proliferation but mainly reflect the sedimentation history and paleo-environment and not a direct effect through warming.

## 1 Introduction

Temperatures in high-latitude regions have been rising twice as fast as the global average over the last 30 years (IPCC in Climate Change 2013, 2013) and are predicted to experience the globally strongest increase in the future (IPCC in Climate Change 2013, 2013; Kattsov et al., 2005). In the northern hemisphere, 24 % of the land surface (Zhang et al., 2003) and large areas of the Arctic shelves are underlain by permafrost (Brown et al., 1997). With 1672 Pg carbon (Schuur et al., 2008), the northern circumpolar permafrost zone stores about twice as much carbon as currently found in the atmosphere (Schuur et al., 2009; Zimov et al., 2006). About 88% of this carbon occurs in permafrost soils and deposits (Tarnocai et al., 2009). Permafrost harbours numerous ancient but viable cells (Bischoff et al., 2013; Gilichinsky et al., 2008; Graham et al., 2012; Koch et al., 2009; Mackelprang et al., 2011; Wagner et al., 2007) that can remain active at extremely low temperatures (Hultman et al., 2015; Rivkina et al., 2000). With increasing permafrost age, microbial communities show adaptations to the permafrost biophysical environment and specialize towards long-term survival strategies such as increased dormancy, DNA repair or stress response (Johnson et al., 2007; Mackelprang et al., 2017). Following the trend of air temperature increase in the northern hemisphere, continuous permafrost warmed by about 0.3°C over the last decade at a global scale (Biskaborn et al., 2019). Warming of permafrost can substantially increase liquid water content, sediment diffusivity and permeability (Overduin et al., 2008; Rivkina et al., 2000; Watanabe and Mizoguchi, 2002) potentially mobilizing carbon in the form of trapped methane (Portnov et al., 2013; Shakhova et al., 2010, 2014; Thornton et al., 2016). Microbial community composition was reported to be responsive to temperature changes (Luo et al., 2014; Rui et al., 2015; Weedon et al., 2012; Xu et al., 2015; Zhang et al., 2005; Zogg et al., 1997). However, results on the extent of these community changes and their dependence on exposure time are contradictory (Allison et al., 2010; Schindlbacher et al., 2011; Walker et al., 2018; Weedon et al., 2017; Xiong et al., 2014; Zhang et al., 2016). In general, the microbial community response to warming appears to be delayed (DeAngelis et al., 2015) and the effect of warming might take decades to affect the microbial community composition (Radujković et al., 2018; Rinnan et al., 2007). Not only microbial community composition can be responsive to temperature but also microbial abundance especially in systems with weak energy constraints. Microbial abundance correlates with enzymatic activities and methane production (Taylor et al., 2002; Waldrop et al., 2010), which are sensitive to temperature. Microbial growth, respiration and carbon uptake can correlate with microbial biomass (Walker et al., 2018). Thus, substantial permafrost warming on long time-scales could affect microbial community composition and abundance before permafrost thaws.

Submarine permafrost provides an analogue for rising permafrost temperatures over time-scales of centuries and millennia. Submarine permafrost of the Arctic Sea shelves originally formed under terrestrial (subaerial) conditions and was inundated by post-glacial sea level rise during the Holocene (Romanovskii and Hubberten, 2001). Upon sea transgression, permafrost degraded over thousands of years as the relatively warm ocean water warmed the submerged sea floor. Mean annual bottom water temperatures in the Laptev Sea (East Siberian Arctic shelf) are 12 to 17 °C warmer than the annual average surface temperature of terrestrial permafrost (Romanovskii et al., 2005). Even today, new submarine permafrost is created by erosion of Arctic permafrost coasts (Fritz et al., 2017), which account for 34% of the coasts worldwide (Lantuit et al., 2012). In a recent study, we compared submarine sediment cores from two locations on the Siberian Arctic Shelf and looked at the combined effect of permafrost inundation time and seawater intrusion on microbial communities. We showed that flooding by sea water reduced permafrost bacterial abundance and changed bacterial community composition due to the penetration of seawater into a former freshwater habitat (Mitzscherling et al., 2017). It was suggested that in addition to the effect of seawater infiltration, the sediment warming taking place over millennia could lead to proliferation. However, the specific effect of long-term permafrost warming independent of thawing has not been assessed so far. Here we hypothesize that millennial-scale permafrost warming directly increases microbial abundance and alters microbial community composition. We used submarine permafrost sediments of comparable age and physicochemical properties that differed in temperature by more than 10 °C due to different periods of inundation and sediment warming and assessed total microbial and bacterial abundances and community composition relative to temperature, pore-water chemistry and sedimentation history.

## 2 Materials and Methods

### 2.1 Study site and drilling

The study area (~73°60'N, 117°18'E) is situated in the western part of the Laptev Sea, on the East Siberian Arctic Shelf (Fig. 1). Mean annual bottom water temperatures in the Laptev Sea range between -1.8 °C to -1 °C (Wegner et al., 2005) leading to sediment temperatures of -1.0 °C and -2.0 °C within the largest part of the shelf (Romanovskii et al., 2004). We investigated four cores (C1-C4, Fig. 2a) that were retrieved along an onshore-offshore transect in the coastal region of Cape Mamontov Klyk in 2005 (Overduin, 2007; Rachold et al., 2007). Cores were named after the order of drilling and we kept this order (C1, C4, C3, C2) for better comparability with previous studies (Koch et al., 2009; Mitzscherling et al., 2017; Overduin et al., 2008; Winkel et al., 2018). From onshore to offshore all cores were characterized by an increase in water depth, in depth to the ice-bonded permafrost table (Fig. 2a, Table S1) and in ground temperature (Table S2) (Overduin, 2007; Rachold et al., 2007). The transect was characterized by a temperature gradient that covered an increment of more than 10 °C compared to the onshore permafrost. Thereby, each core displayed its own unique temperature range (Fig. 2b).

Assuming a constant mean annual coastal erosion rate of 4.5 m yr$^{-1}$ (Grigoriev, 2008) the drill site located furthest offshore (C2, 11.5 km off the coast) was inundated approximately 2500 years ago (Rachold et al., 2007). Accordingly, the drill sites C3 and C4, located 3 km and 1 km off the coast, were inundated around 660 and 220 years ago, respectively. More recent analysis based on remote sensing shows that 40-year coastal erosion rates for the same stretch of coastline between 1965 and 2007 were slower (about 2.9 m yr$^{-1}$) (Günther et al., 2013), which would translate into even longer inundation periods. However, in the present study we refer to Grigoriev (2008), which are based on direct observations of coastal erosion at the C1 coring site. Drilling was performed with a hydraulic rotary-pressure system (Drilling Technologies Factory, St. Petersburg, Russia, Model URB-2A-2) and without the use of any drilling fluid. All samples were frozen immediately after recovery and were kept at -22 °C until further processing. Temperature measurements at all sites were done using thermistors and infra-red sensors (Junker et al., 2008).

## 2.2 Sample selection

Each of the four drill cores exhibited different sedimentological units. Lithostratigraphic Unit II was identified in all cores (Fig. 2a) and was entirely located within the ice-bonded permafrost. Irrespective of the permafrost temperature Unit II sediments of all cores were cemented mainly by pore ice but were also characterized by terrestrial permafrost features like ice lenses, ice veins and ice-wedges. Photographs of (Winterfeld et al., 2011) show similar ice and sediment structures of the terrestrial core C1 and the outermost submarine core C2. Depth location of Unit II within each core can be found in Table S1. This unit was deposited during the late Pleistocene, was warmed without thawing, and had so far remained unaffected of seawater infiltration. On the basis of a PCA analysis (see next chapter and Fig. 3) and previous lithostratigraphic descriptions (Winterfeld et al., 2011) all further analysis was conducted on samples from Unit II. The ages of the sediment are published in Winterfeld et al. (2011). The present study refers to sediment ages determined by optically stimulated luminescence (OSL) on quartz and infrared optically stimulated luminescence (IR-OSL) on feldspars. OSL ages of Unit II sediments from core C1 range from 30.5 ± 2.0 ka at 22 m below surface (m bs) to 114 ± 6 ka at 50 m bs. OSL ages range from 97 ± 6 to 112 ± 8 ka between 23 and 30 m below sea floor (m bsf) in core C3 and from 133 ± 8 to 148 ± 14 ka between 37 and 53 m bsf, and increase with depth. IR-OSL ages date back to 59 ± 5.8 ka at around 15 m bsf in C4 and 86 ± 5.9 ka at 44 m bsf and 111 ± 7.5 ka at 77 m bsf in C2. Consequently, sediments of Unit II were deposited during the early to middle Weichselian (Winterfeld et al., 2011).

For molecular analyses we took 6 replicate samples from each of the cores C1 (C1-1 – C1-6), C4 (C4-1 – C4-6) and C3 (C3-1 – C3-6) and 8 replicates from core C2 (C2-1, C2-2, C2-4, C2-5, C2-7, C2-8, C2-9, C2-10) (Fig. 2a). Those replicates were located at different depths within Unit II (Table S4). Samples from C1 were located around 27 to 44 meters below surface, while samples from C4 were taken between 13 and 30 meters below the seafloor, samples from C3 between 9 and 25 m bsf, and samples from C2 between 40 and 58 m bsf. Unit II was mainly composed of sands with varying proportions of silt and to a minor extent of clay, and a frequent occurrence of wood fragments, plant detritus interlayers and small peat inclusions

(Winterfeld et al., 2011). Both, sandy as well as organic-rich deposits were represented by three replicates in C1, C4 and C3 and four replicates in C2 (Table S4). Furthermore, to check for reproducibility we included samples from C2 retrieved in a previous study (Mitzscherling et al., 2017) (sample names CK12xx). In order to prevent contamination caused by the drilling equipment we took the subsamples from the centre of the core. Subsampling was performed in a climate chamber under freezing conditions by using sterile tools. Thus, a contamination of the samples can be excluded.

## 2.3 Pore water and sediment analyses

Pore water of segregated ground ice was extracted from thawed subsamples of the sediment cores using rinsed Rhizons™ (0.15 µm pore diameter). Electrical conductivity, salinity, cation and anion concentrations, stable isotope concentrations ($\delta^{18}$O, $\delta$D), and pH were measured for 183 samples of C1, 67 samples of C2, 38 samples of C3 and 10 samples of C4 in Unit II (Table S3). Electrical conductivity, salinity and pH were measured with a WTW MultiLab 540 using a TetraConTM 325 cell referenced to 20°C. Total dissolved element concentrations ($Ba^{2+}$, $Ca^{2+}$, $K^+$, $Mg^{2+}$, Na+, $Si_{aq}$) were determined by inductively coupled plasma optical emission spectrometry (ICP-OES, Optima 3000XL, Perkin-Elmer, Waltham) (Boss and Frieden, 1989). Dissolved anion concentrations ($Cl^-$, $SO_4^{2-}$, $Br^-$, $NO_3^-$) were measured using a KOH eluent and a latex particle separation column on a Dionex DX-320 ion chromatographer (Weiss, 2001). The pore water stable isotopes ($\delta$D and $\delta^{18}$O) of segregated ground ice were determined following (Meyer et al., 2000) using a Finnigan MAT Delta-S mass spectrometer in combination with two equilibration units (MS Analysetechnik, Berlin).

Dissolved organic carbon (DOC) was measured as non-purgeable organic carbon via catalytic combustion at 680 °C using a Shimadzu TOC-VCPH instrument on samples treated with 20 µl of 30% supra-pure hydrochloric acid. The ice content was determined gravimetrically. Grain sizes were measured with a Coulter LS 200 laser particle size analyzer. The total organic carbon (TOC) was measured with the element analyser VARIO MAX C, while total carbon (TC), total nitrogen (TN) and total sulfur (TS) contents were determined with a CNS analyzer (Elementar Vario EL III).

## 2.4 DNA extraction

Core subsamples were homogenized in liquid nitrogen and DNA was extracted from ~5 g of sediment using a modified protocol of Zhou et al. (1996). The method was described before (Mitzscherling et al., 2017) and in the following we refer to these samples as molecular samples. Quality of the extracted genomic DNA was assessed via gel electrophoresis (Fig. S1). DNA concentration was quantified with the Qubit2 system (Invitrogen, HS-quant DNA) and the crude DNA was purified using the HiYield PCR Clean-Up & Gel-Extraction Kit (SLG) to reduce PCR inhibitors prior to PCR applications.

## 2.5 Quantification of the bacterial 16S rRNA gene

Quantitative PCR was performed using the CFX Connect™ Real-Time PCR Detection System (Bio-Rad Laboratories, Inc.) and the primers S-D-Bact-0341-b-S-17 and S-D-Bact-0517-a-A-18 targeting the bacterial 16S rRNA gene (Table S5). Each

reaction (20 µl) contained 2x concentrate of iTaq™ Universal SYBR® Green Supermix (Bio-Rad Laboratories), 0.5 µM of each the forward and reverse primer, sterile water and 2 µl of template DNA. The qPCR assays comprised the following steps: initial denaturation for 3 min at 95 °C, followed by 40 cycles of denaturation for 3 sec at 95 °C, annealing for 20 sec at 58.5 °C, elongation for 30 sec at 72 °C and a plate read step at 80 °C for 0.3 sec. Melt curve analysis from 65-95 °C with 0.5°C temperature increment per 0.5 sec cycle was conducted at the end of each run. The qPCR assay was calibrated using known amounts of PCR amplified gene fragments from a pure *Escherichia coli* culture. For each sample three technical replicates were analysed and DNA templates were diluted 5- to 100-fold prior to qPCR analysis. The PCR efficiencies based on standard curves were calculated using the BioRad CFX Manager software. They varied between 93 and 99%. All cycle data were collected using the single threshold Cq determination mode.

**2.6 Total cell counts**

Preparation and quantification of the total cell abundance per gram sediment were performed after Llobet-Brossa et al. (1998). The modified protocol was described before by Mitzscherling et al. (2017). Briefly, cells were fixed with 4% paraformaldehyde in phosphate-buffered saline (PBS). After incubation, the sediment was pelleted by centrifugation for 5 min at 9600 g and washed in sterile filtered PBS. Two subsamples of each sample were diluted in PBS and filtered onto a polycarbonate membrane filter (0.2 µm) by applying a vacuum. Total cell counts were determined by SYBR Green I. Fluorescence microscopy was performed with a Leica DM2000 fluorescence microscope using the FI/RH filter cube. A magnification of 100x was used to count cells of either 200 fields of view or until 1000 cells were counted. We counted two filters per sample.

**2.7 High throughput Illumina16S rRNA gene sequencing and analysis**

Sequencing of each sample was performed in two technical replicates. The sequencing primers that were used in this study only target bacteria and comprised different combinations of barcodes (Table S6). PCR amplification was carried out with a T100™ Thermal Cycler (Bio-Rad Laboratories, CA, USA). The PCR mixtures (25 µl) contained 1.25 U of OptiTaq DNA Polymerase (Roboklon), 10x concentrate buffer C (Roboklon), 0.5 µM of the sequencing primers S-D-Bact-0341-b-S-17 and S-D-Bact-0785-a-A-21 (Table S5), dNTP mix (0.2 mM each), additional 0.5 mM of $MgCl_2$ (Roboklon), PCR-grade water, and 2.5 µl of template DNA. PCR conditions comprised an initial denaturation at 95°C for 5 min, followed by 35 cycles of denaturation (95°C for 30 s), annealing (56°C for 30 s) and elongation (72°C for 1 min), and a final extension step of 72°C for 10 min. The PCR products were purified from agarose gel with the HiYieldPCR Clean-Up and Gel-Extraction Kit (Südlabor, Gauting, Germany) and were quantified with the QBIT2 system (Invitrogen, HS-Quant DNA). They were mixed in equimolar amounts and sequenced from both directions (GATC Biotech, Konstanz) based on the Illumina MiSeq

technology. The library was prepared with the MiSeq Reagent Kit V3 for 2×300 bp paired-end reads. The 15% PhiX control v3 library was used for better performance due to different sequencing length.

## 2.8 Sequence analysis and bioinformatics

The data analysis of raw bacterial sequences started with the quality control of the sequencing library by the tool FastQC (Quality Control tool for High Throughput Sequence Data http://www.bioinformatics.babraham.ac.uk/projects/fastqc/ by S. Andrews). The tool CutAdapt [Martin, 2011] was used to demultiplex the sequence reads according to their barcodes and to subsequently remove the barcodes. Forward and reverse sequenced fragments with overlapping sequence regions were merged using PEAR [J. Zhang et al., 2014], and the nucleotide sequence orientation was standardized. Low-quality sequences were filtered and trimmed by Trimmomatic [Bolger et al., 2014], and chimeras were removed by Chimera. Slayer. Finally, the QIIME pipeline was used to cluster sequences into operational taxonomic units (OTUs) and to taxonomically assign them employing the SILVA database (release 123) with a cutoff value of 97% [Caporaso et al., 2010].

## 2.9 Statistics

Prior to statistical analysis, absolute singletons and $OTU_{0.03}$ (operational taxonomic units of clustered sequences with 97% similarity level) not classified as bacteria or classified as chloroplasts or mitochondria were removed. In addition, $OTU_{0.03}$ with reads <0.5% of total read counts in each sample were removed to reduce background noise. The background noise was estimated with the help of a positive control (*E. coli*), where the number of OTUs is known prior to sequencing. Absolute read counts were transformed into relative abundances in order to standardize the data and to make technical replicates comparable. Relative abundances of technical replicates were merged to mean relative abundances for bacterial community analysis i.e. the bubble plot and CCA. Samples having < 15.000 raw reads were checked for divergent relative abundances within duplicates (Table S7) and excluded from the calculation of mean relative abundances when the discrepancy was too big. Variation in $OTU_{0.03}$ composition, 16S rRNA gene and total cell abundance between samples and among drill sites, as well as correlations of the abundance and $OTU_{0.03}$ composition with environmental parameters were assessed using the Past 3.14 software (Hammer et al., 2001) and R, especially the vegan and MASS packages. Principal component analyses (PCA) based on Euclidean distance were used to assess variation in environmental variables across the different sediment units and within Unit II. Prior to analysis, all environmental data were standardized by subtracting the mean and dividing by standard deviation. To assess the correlations of bacterial and microbial abundance with environmental parameters the rank-based Spearman correlation was calculated. The Bray-Curtis dissimilarity was used to assess the beta diversity of the microbial communities in an NMDS plot. Environmental factors that might influence its composition were determined by an environmental fit into the ordination. The significance of the variance introduced by the identified environmental factors was tested using a permutational approach as implemented in the adonis function of the vegan package. Factors were tested for

auto-correlation as implemented in the corrplot package. A linear model of the remaining factors was subject to a redundancy analysis which was tested for signficance using the analysis of variance (ANOVA). ANOVA and the Tukey's pairwise post-hoc test were conducted to test whether DOC concentrations of the cores differed.

## 3 Results

### 3.1 Physicochemical pore water and sediment properties

Temperature (Fig. 2b) of Unit II was lowest in the terrestrial borehole (C1, constantly at around -12.4 °C at the time of drilling (Junker et al., 2008) and between -12.0 and -12.5 °C recently measured over a 2 year period (Kneier et al., 2018)) and increased with distance to the shore. According to (Junker et al., 2008) C4 exhibited a temperature range from -7.1 to -5.8 °C. Ground temperatures of C3 and C2 were similar with mean values of -1.4 and -1.5 °C, respectively, and showed marginal variation. C3 exhibited a slightly higher mean temperature than the longest inundated core C2.

Overall, the salinity of Unit II was low (Fig. 2b, (Winterfeld et al., 2011)). In C4, the drill site located closest to the coast, Unit II had the highest pore water salinity (mean = 5.6 PSU) ranging from 0.9 to 17.6 PSU (Table S2), which spans freshwater to mesohaline water but is much below seawater salinities. In comparison, bottom-water salinities at the drill sites ranged between 29.2 and 32.2 PSU (Overduin et al., 2008). Salinity in C3 reached a mean value of 1.1 PSU. The submarine core furthest offshore (C2) and the terrestrial core (C1) had a mean pore water salinity of around 0.8 and 0.5 PSU, respectively. The stable isotopes $\delta D$ and $\delta^{18}O$ of the sediment cores C1 and C4 exhibited similar mean values of -22 ‰ for $\delta^{18}O$ and around -178 ‰ for $\delta D$, albeit a greater variance in C1 (Fig. 2c, Table S2). Sediments of C3 were characterized by higher and constant isotope values of around -20 ‰ for $\delta^{18}O$ and -158 ‰ for $\delta D$. In core C2, the isotope values were smaller with mean values of -28 ‰ for $\delta^{18}O$ and -213 ‰ for $\delta D$ (Table S2).

DOC concentrations were lowest in Unit II of core C2, the core furthest offshore, and ranged from 4 to 41 mg C L$^{-1}$, with a mean value of 17 mg C L$^{-1}$ (Fig. S2). Towards the coast the DOC content increased to mean values of 43 mg C L$^{-1}$ in C3 and 96 mg C L$^{-1}$ in C4. The terrestrial core C1 had a mean DOC concentration of around 48 mg C L$^{-1}$ with values ranging from 4 to 305 mg C L$^{-1}$, thereby having by far the highest measured DOC concentration of all cores. The TOC content in this Unit II was generally very low with mostly < 0.5 wt%. While C1 and C4 had lowest mean values of 0.17 wt%, the TOC content increased with distance to the coast to 0.22 wt% in C3 and 0.33 wt% in C2 (Table S3). The pH of Unit II sediments ranged from slightly acidic to slightly alkaline values. In cores C1 and C4 the pH ranged from 5 to 7.9, whereas values of C2 and C3 were higher ranging from pH 6.5 to 8.0. Mean pH values of all cores were around pH 7 to 7.5. Other pore water data like anion and cation concentrations, conductivity, CNS, grain sizes and the gravimetrically determined water content can be found in Table S3.

All environmental, sedimentological and pore water data (Table S3) were used to conduct principal component analyses (PCA) to check for the level of similarity within Unit II. Unit II formed a dense cluster relative to the other sediment units

(Fig. 3 Insert). Focusing on samples from Unit II only (Fig. 3) confirmed highly similar physicochemical characteristics of this unit in all cores even though C2 and C3 clustered along the axis PC2, while C1 and C4 were more randomly scattered. Variance between samples was mainly explained by grain sizes, pore water stable isotope concentrations and to a lesser extent by pH.

## 3.2 Microbial abundance

Overall microbial abundance decreased from onshore to offshore (C1, C4, C3) and had increased again in the drill site located furthest from the coast (C2). The terrestrial permafrost core C1 and the submarine core C2 had highest DNA concentrations (Fig. S3), total cell counts (TCC) (Fig. 4a) and bacterial 16S rRNA gene copy numbers of all cores (Fig. 4b). Lowest DNA concentrations and TCC were observed in core C3, whereas lowest numbers of bacterial 16S rRNA gene copies were found in core C4. All three abundance measures (DNA concentrations, TCC, and bacterial 16S rRNA gene copy numbers) significantly correlated with each other (Table S8). DNA concentrations reached mean values of 141.6 ng $g^{-1}$ and 106.9 ng $g^{-1}$ in C1 and C2, respectively, whereas the mean DNA concentration in C4 and C3 were 88.5 and 19.8 ng $g^{-1}$ (Table S9). Mean TCC reached a value of 5 x $10^7$ $g^{-1}$ in C1. C4 and C2 had similar values of 1.3 x $10^7$ $g^{-1}$ and 1.5 x $10^7$ $g^{-1}$, while cell numbers of C3 were one order of magnitude lower (1.5 x $10^6$ $g^{-1}$). Bacterial 16S rRNA gene copy numbers usually exceeded TCC by an order of magnitude, with mean values of 1.6 x $10^8$ $g^{-1}$ and 2.9 x $10^8$ $g^{-1}$ in C1 and C2, but lower mean values of 3.6 x $10^7$ $g^{-1}$ and 1.7 x $10^7$ $g^{-1}$ in C4 and C3, respectively.

A correlation analysis (Table 1) revealed that microbial and bacterial abundance measures including DNA concentrations, 16S rRNA bacterial gene copies and TCC correlated with each other (Fig. 4c). They further showed a significant rank-based negative correlation with salinity ($p < 0.05$, Spearman $-0.63 \leq r_s \leq -0.35$), cations ($K^+$, $Mg^{2+}$, $Na^+$) and anions ($Cl^-$, $Br^-$) ($p < 0.05$, $-0.71 \leq r_s \leq -0.39$), and $\delta^{18}O$ ($p<0.05$, $-0.38 \leq r_s \leq -0.37$). Furthermore, DNA concentrations negatively correlated with temperature ($p < 0.05$, $r_s = -0.37$) and pH ($p < 0.05$, $r_s = -0.44$), while TCC negatively correlated with temperature ($p < 0.01$, $r_s = -0.64$) and 16S rRNA gene copies with pH ($p < 0.01$, $r_s = -0.24$). Positive correlations were found for DNA and 16S rRNA gene copies with total organic carbon (TOC, $p < 0.05$, $r_s > 0.34$) and the water content ($p < 0.01$, $r_s = 0.47$).

## 3.3 Bacterial community composition

The most abundant bacterial taxa were Actinobacteria (class), Chloroflexi (Gitt-GS-136, KD4-96), Clostridia (class), Gemmatimonadetes, and Proteobacteria (primarily Alpha- and Betaproteobacteria) (Fig. 5). *Candidatus* Aminicenantes (candidate phylum OP8) and *Candidatus* Atribacteria (candidate phylum OP9) were highly abundant in core C3, where Actinobacteria, Chloroflexi, and Gemmatimonadetes were almost absent.

Grouping patterns of the bacterial community based on the $OTU_{0.03}$ composition of the samples and the Bray-Curtis dissimilarity were visualized using a non-metric multidimensional scaling (NMDS, Fig. 5). The NMDS showed a clustering of samples according to their borehole location for C2 and C3, while communities of C1 and C4 were more scattered. We

fitted environmental gradients with the NMDS ordination in order to test for correlation between the bacterial community compositions at each drill site with environmental parameters ($p < 0.05$). Samples located at the bottom left of the plot originated from a greater depth (C1 and C2) than samples to the top right (C3 and C4). Variance of samples from the bottom to the top was explained by rising pH, permafrost temperature and total sulphur content, while variance of samples from the left to the right side are likely explained by increasing values of $Ba^{2+}$ and the pore water stable isotopes $\delta^{18}O$ and $\delta D$ - a proxy for paleo-temperature and –climate. The bacterial community of C3 was most distinct and clustered furthest from communities of all other sites. It was linked with the pore water stable isotopes $\delta^{18}O$ and $\delta D$, $Ba^{2+}$ and the sample depth. The variance between C1, C4 and C2 samples are explained by permafrost temperature differences across the cores (Fig. 2b). A subsequent permutational analysis of variance showed that depth, temperature, pH, TS, $\delta D$, $\delta^{18}O$, and $Ba^{2+}$ contribute to the variance in the microbial community composition (Table S11), whereof $\delta^{18}O$ and $\delta D$ show a high auto-correlation. A redundancy analysis showed that the explanatory variables depth, temperature, pH and $\delta^{18}O$ significantly explain parts of the variance in the microbial composition ($p = 0.001$).

Despite the overlaps within the CCA ordination, a one-way PerMANOVA revealed that the variance between each of the clusters was significantly higher than within single clusters (Table S12), i.e., the bacterial subpopulations of each drill site were significantly different from each other.

## 4 Discussion

The present study aimed at understanding the effect of long-term permafrost warming independent of thaw on microbial community composition and abundance. The observed significant negative rank-based correlation between increasing temperature and total cell counts (TCC) contradicts our hypothesis that millennial-scale permafrost warming directly increases microbial abundance. It is, however, in line with related studies on arctic and subarctic soil microbial communities where a negative effect of increasing temperature on microbial abundance was assigned to freeze-thaw cycles (Schimel et al., 2007; Skogland et al., 1988) and substrate depletion (Walker et al., 2018). Both effects are, however, unlikely here. Firstly, sample depths were always more than 10 m below surface and sea floor, respectively, and freeze-thaw cycles within the investigated Unit II can be excluded. Secondly, preservation, rather than depletion, of substrates was more likely in the two submarine cores C3 and C4, where DOC contents were comparable to that of the cold terrestrial permafrost of C1 (Fig. S2). The degradation of DOC can be used as measure for microbial carbon turnover (Seto and Yanagiya, 1983) and the DOC concentration usually correlates with microbial abundance (Junge et al., 2004; Smolander and Kitunen, 2002; Vetter et al., 2010). The cores C3 and C4 had significantly lower TCC and bacterial gene copy numbers ($10^6$ cells and $10^5$ gene copies) than the onshore core C1 and the C2 core furthest offshore ($10^7$ cells and $10^6$ gene copies). Thus, microbial activity and substrate utilization were likely low in C3 and C4. A negative influence of permafrost warming on microbial abundance is further challenged through some indication for microbial proliferation in core C2, which had experienced longest warming of

all cores. In detail, TCC in C2 were higher than in the other submarine cores while DOC values were lower in C2, significantly different from C4 and C1 (Table S13). Permafrost warming for more than two millennia may have enabled microbial communities to adapt to the new temperature regime and sediment properties as suggested before (Mitzscherling et al., 2017). Besides permafrost warming changing pore-water salinity had an effect on the microbial abundance. Rising permafrost temperature strongly correlates with TCC whereas salinity correlates strongest with bacterial gene copy numbers (Table 1). Bacterial 16S rRNA gene copy numbers were lowest in core C4 ($10^5$ gene copies), where pore-water salinities were elevated (electrical conductivity values >2000 µS cm$^{-1}$, Table S3). Low gene copy numbers ($10^5$ gene copies) may result from osmotic stress that limits microbial growth (Galinski, 1995; Rousk et al., 2011) and decreases microbial abundance in sediments (Jiang et al., 2007; Rath and Rousk, 2015; Rietz and Haynes, 2003; Wen et al., 2018). We argue that the different levels of salinity are relicts of the paleo-climate and varying landscape types (e.g. thermokarst lakes and lagoons, fluvial, floodplain, Fig. S5 and Table S14) that formed Unit II during the last glacial cycle, i.e. the Weichselian glaciation 117 – 10 ka BP (Svendsen et al., 2004). According to the IR-OSL ages Unit II of C4 was deposited ~60 ka BP and earlier. Conductivity values in C4 that were higher than 2000 µS cm$^{-1}$ could be the result of strong evaporation. The climate in the Laptev Sea region during the middle Weichselian (75 – 25 ka BP) was of extremely continental type characterized by low precipitation throughout the year and relative warm summers (Hubberten et al., 2004). Also, salinity values in Unit II of core C4 are lower than in the seafloor sediments of the same core but higher than in the sediment layer in between (Fig. 2b), supporting the idea that differences in salinity reflect the paleo-environment and climate, and not an infiltration of seawater during the Holocene transgression. The presence of a temporary shallow thermokarst lake at the drilling site of C4 and following summer evaporation is one possible scenario leading to elevated salt concentrations (Larry Lopez et al., 2007). A strong influence of the paleo-climate on recent microbial abundance is further supported through a significant correlation between microbial abundance with δ$^{18}$O values (Table 1). The stable isotope composition of ground ice is widely used as an archive for paleo-climatic information and for the determination of ground ice genesis (Meyer et al., 2002a, 2002b; Vasil'chuk, 1991). Compared to the other cores, C3 for example was enriched in heavy isotope species of δ$^{18}$O (-20 to -15‰) and δD (-150 to -160‰), suggesting warmer temperatures at the time of deposition (Meyer et al., 2002b). As ground ice is mainly fed by summer and winter precipitation, its isotopic composition reflects the annual range of air temperatures. Isotope changes towards heavier values could also be the result of larger amounts of summer rain as well as less winter snow preserved in the ice. Assuming that IR-OSL ages of Winterfeld et al. (2011) are correct, sediments of C3 were deposited at around 50 ka BP and later. Thus, C3 sediments were probably deposited during a period where the extremely dry continental climate with relatively warm summers was especially pronounced (between 45 and 35 ka BP) (Hubberten et al., 2004).

We suggest that microbial community composition like microbial abundance reflects the paleoclimate and sedimentation history and not a direct effect of permafrost warming. In detail, we observed a weak correlation between community composition with permafrost temperature and a strong correlation with pore water stable isotope values and depth, i.e. age. This suggestion is supported by similar findings in sea sediments as well as in lacustrine sediments. Microbial taxa of

Arabian Sea sediments reflected past depositional conditions and exhibited paleo-environmental selection (Orsi 2017), while the microbial population in sediments of Laguna Potrok Aike in Argentina changed in response to both past environmental conditions and geochemical changes during burial (Vuillemin, 2018). The microbial communities in core C3 which were most distinct from the other locations (Fig. 6) may thus reflect the higher paleo-temperatures and different proportions of summer and winter precipitation discussed earlier. The strongest correlation of the bacterial community composition was, however, found with pH. Soil pH is a major factor controlling the bacterial diversity, richness and community composition on a continental scale (Fierer and Jackson, 2006; Lauber et al., 2009). On a global scale pH is also one of the major controls of archaeal communities (Wen et al., 2017). Fierer and Jackson (2006) showed that the richness and diversity of bacterial communities differed between ecosystem types, which could be explained by pH. This substantiates our suggestion that Unit II and the bacterial community therein was formed under different paleo-climatic conditions and varying landscape types during the last glacial cycle. However, the limited number of environmental samples and the inference of other correlating environmental factors might decrease the statistical powers to see a more significant effect of temperature on the microbial community.

Independent of core C3, microbial community composition showed substantial site specific differences. This local scale variation in community composition (β-diversity) likely results from the distance between the coring sites because β-diversity increases with increasing distance when environmental conditions differ (Lindström and Langenheder, 2012) and when dispersal is as limited as it is in permafrost environments (Bottos et al., 2018). Our data suggest that the bacterial community in submarine permafrost sediments has experienced a weak selection after deposition and mostly reflects the paleo-environmental and climatic conditions. Thereby this study joins a number of other studies reporting on microbial groups that are referred to as "the paleome". Those studies found correlations between the microbial diversity and past depositional conditions (Lyra et al., 2013; Orsi et al., 2017; Vuillemin et al., 2016). Marine communities were found in terrestrial settings or soil communities in (sub)seafloor sediments (Ciobanu et al., 2012; Inagaki et al., 2015; Inagaki and Nealson, 2006). Like those, our study implies that the bacterial communities in permafrost soils under the seafloor underwent a weak selection pressure after burial either through dormancy or very low generation times under freezing conditions.

Irrespective of the effect of permafrost warming on microbial community composition and abundance, the cell counts and microbial taxa of this study expand our knowledge about microbial life in permafrost. The bacterial taxa dominating in the submarine permafrost samples were amongst the phyla that commonly occur in Artic permafrost and the active layer, like Proteobacteria, Firmicutes, Chloroflexi, Acidobacteria, Actinobacteria and Bacteroidetes (Jansson and Taş, 2014; Liebner et al., 2009; Mitzscherling et al., 2017; Taş et al., 2018). Furthermore, the most abundant taxa Actinobacteria, Chloroflexi, Firmicutes, Gemmatimonadetes and Proteobacteria (Fig. 5) are amongst the groups that were found to be active under frozen conditions in permafrost (Coolen and Orsi, 2015; Tuorto et al., 2014). The non-spore forming Actinobacteria were reported to dominate permafrost since they are well adapted to freezing conditions (Johnson et al., 2007). They are metabolically active at low temperatures and possess DNA-repair mechanisms. Firmicutes and Proteobacteria likely resist long-term

exposure to subzero temperatures as they take advantage of nutrient and water availability (Johnson et al., 2007; Yergeau et al., 2010). In addition, many members of the Firmicutes are able to form spores. Candidatus Atribacteria, which dominated in the core C3, were recently described to harbor functions for survival under extreme conditions like high salinities and cold temperatures (Glass et al., 2019). They are further one of the cosmopolitan groups in the subseafloor and dominate the bacterial community in deep anoxic sediments with low organic carbon contents (Orsi, 2018). This makes Atribacteria another candidate for activity under in situ conditions in submarine permafrost. Genome-based metabolic prediction shows that Ca. Atribacteria can ferment sugars and propionate producing $H_2$, which is a critical source of energy in anoxic settings, and they have the potential to polymerize carbohydrates and store them in shell proteins of bacterial microcompartments, thus increasing their fitness and leading to their selection (Orsi, 2018). Besides subseafloor sediments Ca. Atribacteria were found to be abundant in lacustrine sediments in Argentina that were deposited under similar environmental conditions like C3, with permafrost and reduced vegetation in the catchment, an active hydrology reworking and dispersing the soils, and a very low organic carbon content. Also climatic conditions in the sedimentation period of the lacustrine sediments were similar to that of Unit II in C3, covering the driest period of the record and overall positive temperatures (Vuillemin et al., 2018).

The TCC of the onshore permafrost core C1 were in the upper range of cell counts ($10^6$-$10^7$ cells $g^{-1}$) reported for other permafrost environments (Gilichinsky et al., 2008; Jansson and Taş, 2014; Steven et al., 2006) and TCC of the three submarine permafrost cores were comparable to microbial abundances from organic carbon rich sub-seafloor sediments ($10^5$-$10^7$ cells $g^{-1}$) (Kallmeyer et al., 2012; Parkes et al., 2014). TCC and bacterial 16S rRNA gene abundance in cores C1 and C2, which were highest in this study, were at least one order of magnitude lower than values for the active layer, i.e. the seasonally thawed, upper permafrost layer (Kobabe et al., 2004; Liebner et al., 2008, 2015). This is in line with modelling studies on generation times in the subsurface where cells were reported to divide only every ten to hundred years (Jørgensen and Marshall, 2016; Starnawski et al., 2017). It also underlines that the effect of warming on microbial abundance in the investigated submarine permafrost cores was likely poor as discussed earlier. The observation that 16S rRNA gene copies mostly exceeded TCC by an order of magnitude may reflect the long-term preservation of extracellular DNA due to low temperature conditions in permafrost (Stokstad, 2003; Willerslev et al., 2004) and, to a lesser extent, the appearance of multiple 16S rRNA gene copies per cell (Schmidt, 1998). Although qPCR is a good relative quantification method, it is only poorly related to cell counts (Lloyd et al., 2013). In addition, cell counts might be slightly underestimated due to hidden cells below sediment particles (Kallmeyer, 2011).

## 5 Conclusions

Substantial permafrost warming is occurring throughout the Arctic today and the associated response of microbial communities driving the biogeochemical cycling and the formation of greenhouse gases is of general interest. Inundation by

seawater accelerates permafrost warming and results in a steady state of temperature under the present conditions within a few centuries. This makes submarine permafrost a suitable natural laboratory to study the microbial response on climate relevant time-scales. Our results demonstrate that both microbial abundance and community composition even after millennia of submarine permafrost warming by more than 10 °C reflect the paleo-climate and sedimentation history. However, even though we could not finally prove that long-term permafrost warming directly affects microbial abundance and bacterial community composition we found indications for it especially in the core that had experienced longest warming. This deserves more attention, because a direct effect of permafrost warming on microbial abundance, composition and carbon turnover would alter our understanding of the permafrost carbon feedback, which to date only considers permafrost thaw. Based on our work we suggest that future work addresses the responsiveness of microbial communities to permafrost warming through the analysis of organic matter quality (Fischer et al., 2002), chemical composition of permafrost DOM (Spencer et al., 2015; Sun et al., 1997; Ward and Cory, 2015), natural abundance isotope ratios of biomarkers (Boschker and Middelburg, 2002), metagenomics and metatranscriptomics (Coolen and Orsi, 2015; Mackelprang et al., 2017). Finally, in this study the length of the coring transect (~12 km), the age span within and between the cores and hence the comparatively long sedimentation period encompassed by our samples from Unit II had a stronger influence on recent microbial abundance and community than the large level of physicochemical similarity within this unit (Fig. 3 insert). Further studies on the microbial response to permafrost warming should focus on historically more similar samples without neglecting similar physicochemical properties.

**Data availability**

Sequences of the submarine permafrost communities presented in this work were deposited at the NCBI Sequence Read Archive (SRA) with the Project number BioProject ID# PRJNA352907. Bacterial 16S rRNA gene sequences have the sequence read archive accession numbers SRR7908003 - SRR7908028 and are available from Genbank, EMBL, and DDBJ. (https://www.ncbi.nlm.nih.gov/bioproject/PRJNA352907, last access 15 January 2019). Environmental data of the sediment cores are available at https://doi.pangaea.de/10.1594/PANGAEA.895292.

**Author contribution**

SL, DW, MWk and JM formulated the research question and study design. PPO and MNG conducted field work. JM visualized the data and prepared graphs. MWf and PPO provided pore water and physicochemical data. FH conducted the bioinformatics analysis. LM performed and JK supported the cell counting. JM and SL prepared the original draft. All authors contributed to the discussion and interpretation of the data and the writing of the paper.

## Competing Interest

The authors declare that they have no conflict of interest.

## Acknowledgments

Our thanks go to Aleksandr Maslov (SB RAS, Melnikov Permafrost Institute, Yakutsk, Russia), who provided indispensable drilling expertise. We thank Tiksi Hydrobase staff members Viktor Bayderin, Viktor Dobrobaba, Sergey Kamarin, Valery Kulikov, Dmitry Mashkov, Dmitry Melnichenko, Aleksandr Safin, and Aleksandr Shiyan for their field support and Dimitry Yu. Bolshiyanov (Arctic Antarctic Research Institute, St. Petersburg) for logistical issues. Coring was supported by the German Ministry for Education and Research, a Joint Russian German Research Group (HGF-JRG100) of the Helmholtz Association of German Research Centres, and by the EU's INTAS program. Susanne Liebner is grateful for the funding of the Helmholtz Young Investigators Group (grant VH-NG-919). We further thank Anke Saborowski, Antje Eulenburg, Ute Bastian, and Katja Hockun for excellent laboratory support.

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

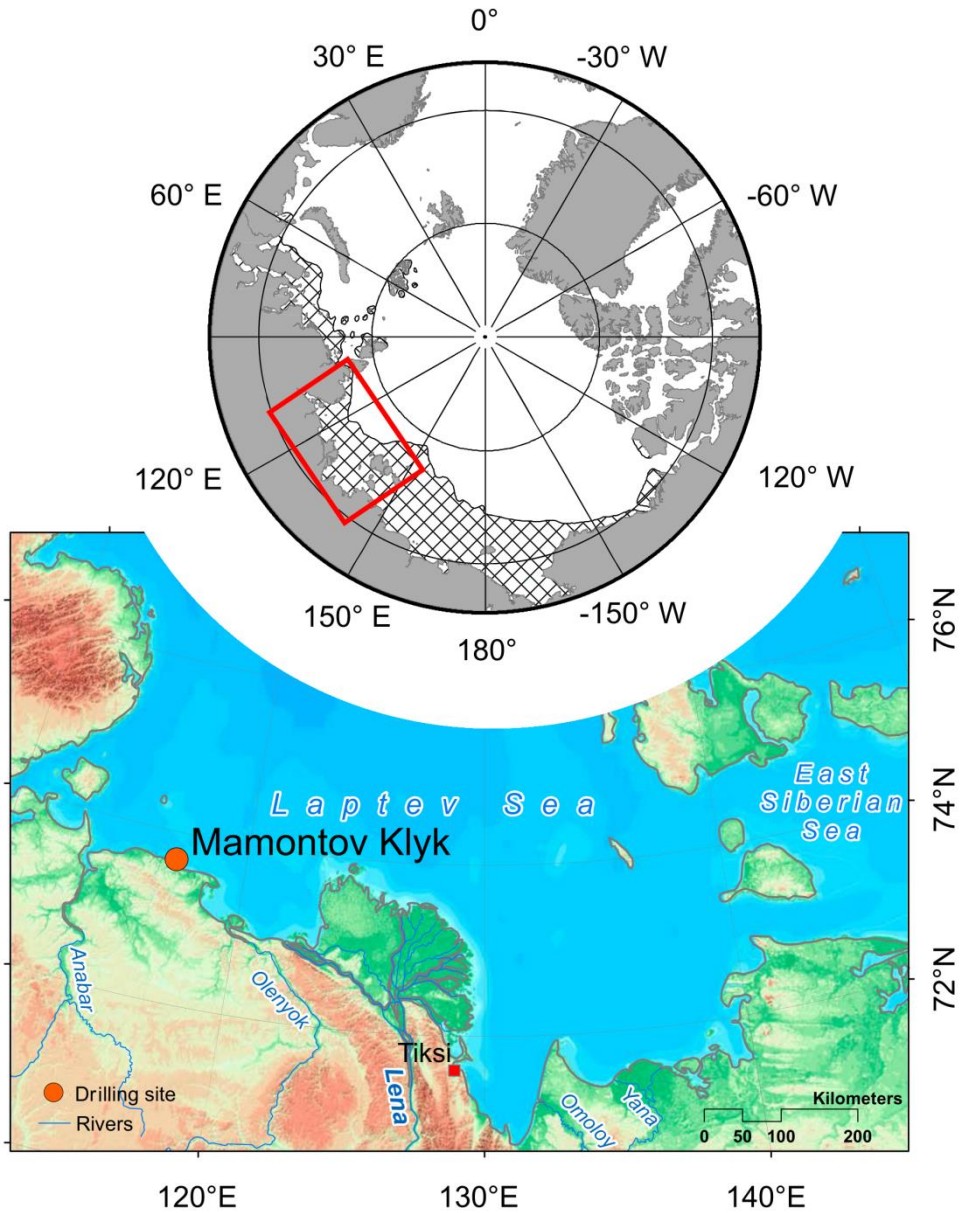

**Figure 1: Geographical location of the study site.** Location of the Laptev Sea on a circumpolar perspective map and the potential extent of submarine permafrost (striped area, based on (Brown et al., 2002)), as well as the geographical location of the drilling site at Cape Mamontov Klyk in the western Laptev Sea. (modified from (Overduin et al., 2015)).

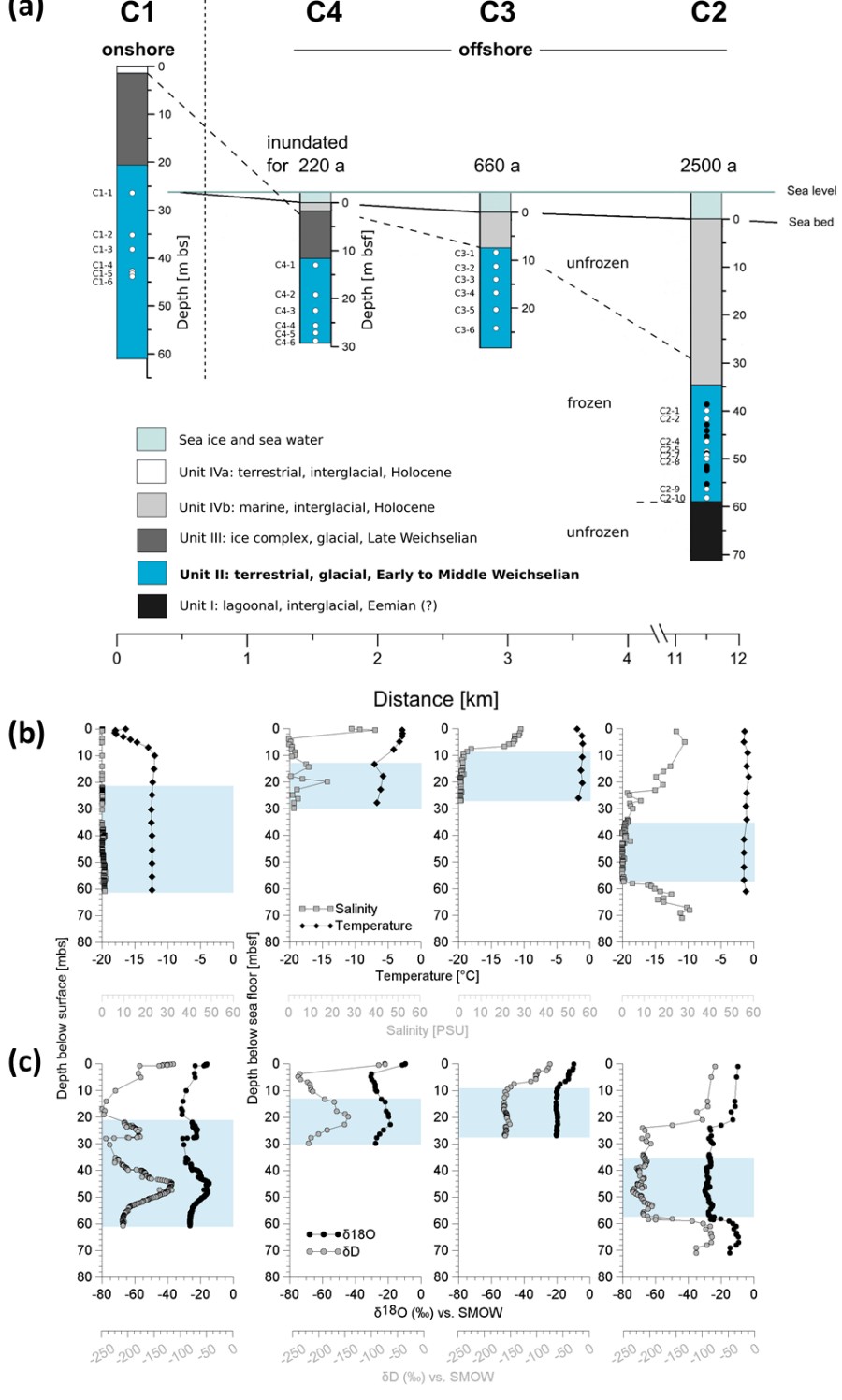

**Figure 2: Overview of the coring transect, position and characteristics of the terrestrial and the submarine sediment cores. a)** Periods of inundation are indicated above each submarine core. Core depth of the terrestrial core is given in m below surface (m bs) and depth of the submarine cores in meters below sea floor (m bsf). The core depths are proportional to each other, whereas the distance scale is only schematic. Affiliation of sediment deposits to discrete sediment units (Unit I - IVb), accumulated under similar environmental conditions in the same glacial or interglacial period, are distinguished by colours. Dots show the depth of the molecular samples. White dots represent samples from this study. Their denomination is indicated to the left. Black dots represent samples from a previous study. **b)** Depth profiles of temperature (black diamonds) and salinity (grey squares) as well as of **c)** the pore water stable isotopes $\delta^{18}O$ (black circles) and $\delta D$ (grey circles) from the cores C1, C4, C3 and C2. The blue shaded area represents Unit II.

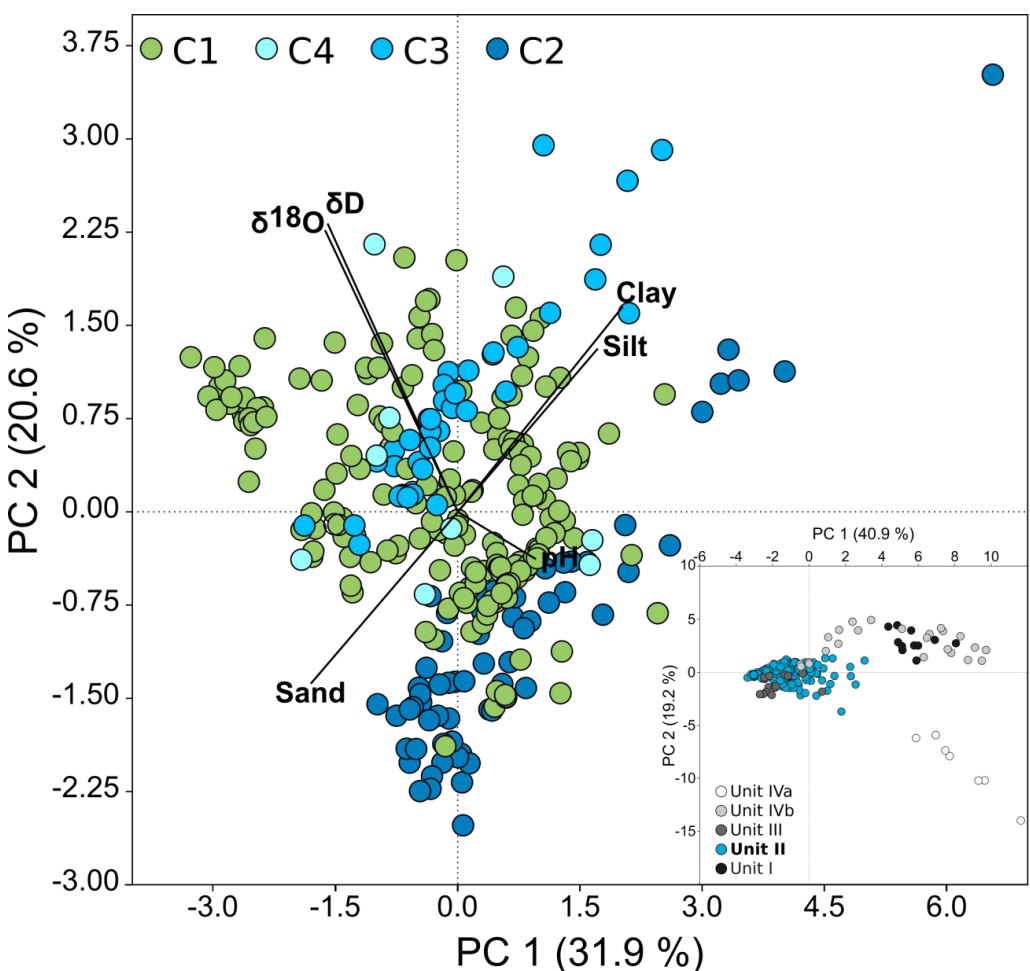

**Figure 3: PCA of environmental, sedimentological and pore water data from Unit II of all four cores** with PC 1 explaining 31.9% and PC 2 explaining 20.6 % of the variance between samples. Vectors show selected physicochemical factors that are mainly responsible for the variance between samples (see loadings plot Fig. S4). C1: n = 183, C2: n = 66, C3: n = 38, C4: n = 9. Outliers located outside the 95% ellipses were removed. The **insert** presents all samples of the onshore-offshore transect coloured irrespective of the cores by Unit (n = 361).

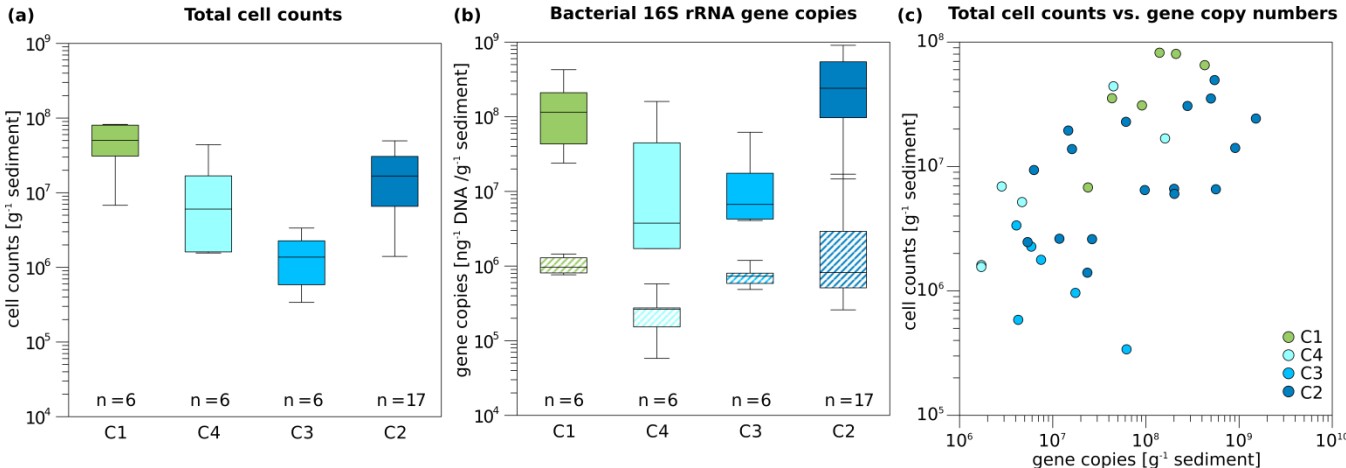

Figure 4: Boxplots of microbial and bacterial abundance in Unit II. a) Total cell counts and b) bacterial 16S rRNA gene copy numbers normalized to gram sediment wet weight (top, solid boxes) and to DNA concentration in ng (bottom, striped boxes) of the cores C1, C4, C3 and C2. Box plots contain the mean values obtained from two technical replicates of cell counts and three technical replicates of 16S rRNA gene copy numbers per biological replicate. Median lines are indicated within the boxes of which the size corresponds to ± 25% of the data, whereas the whiskers show the minimum and maximum of all data. Minimum, maximum and mean values, as well as standard deviation and sample numbers can be found in Table S9. c) Correlation of total cell counts and bacterial 16S gene copy numbers g$^{-1}$ sediment. Strength of the correlation is shown in table 1. Sample points were colored according to drill core.

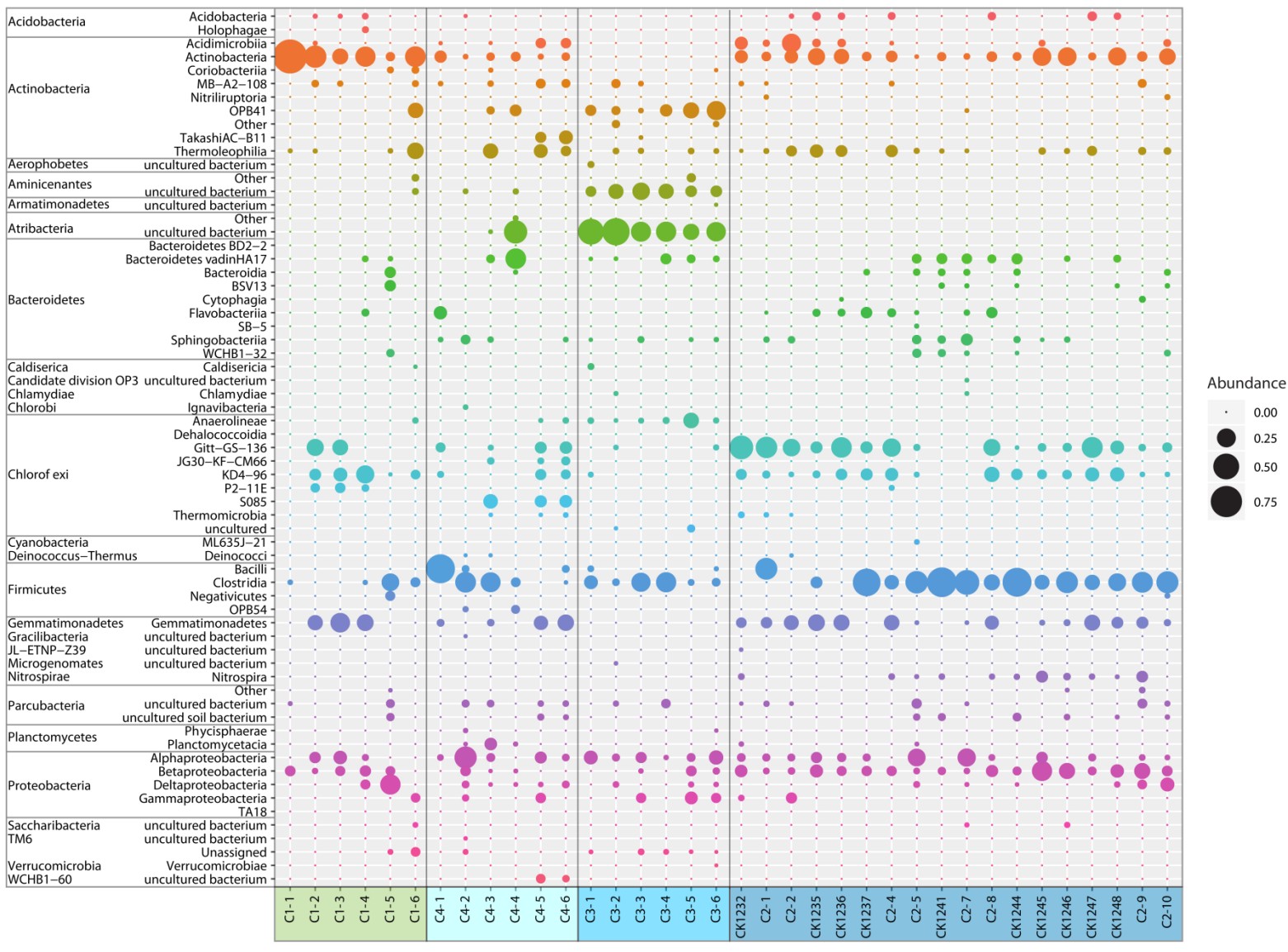

**Figure 5: Relative abundance of bacterial classes from Unit II of the C1 - C4 cores.** Coloured boxes and sample names below indicate the particular core. Sample names were explained earlier. Bubbles represent the mean value of relative abundances from two technical replicates.

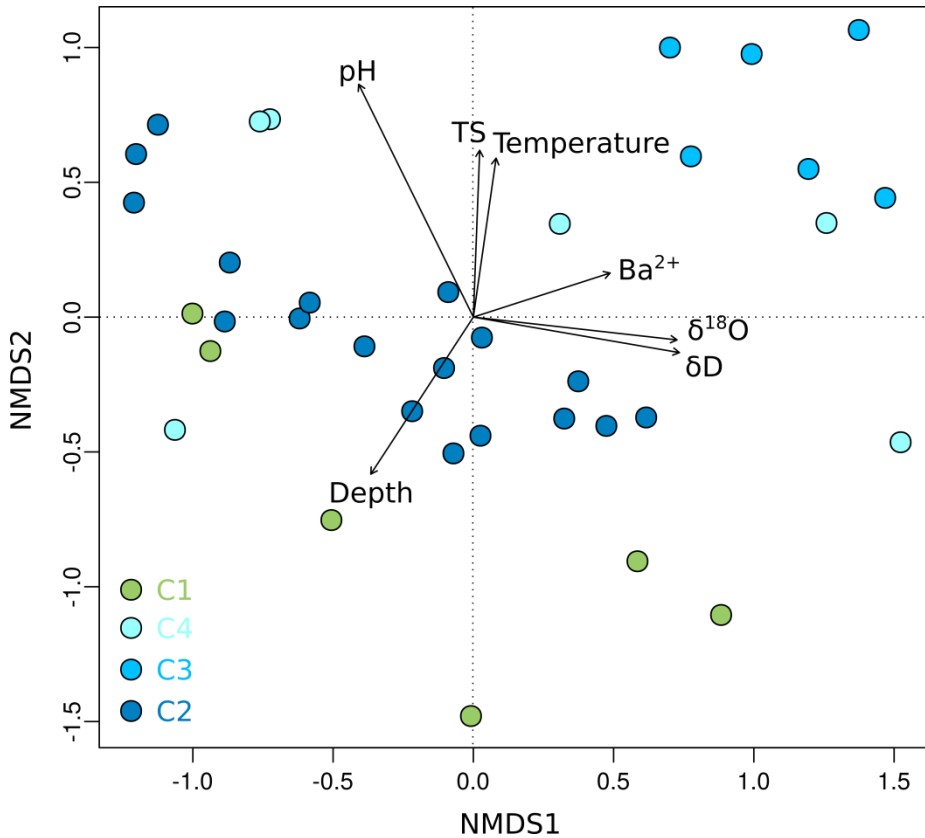

**Figure 6: Non-metric multidimensional scaling (NMDS) plot of OTU$_{0.03}$ data from Unit II in dependence on environmental parameters.** Shown are environmental factors that contribute significantly ($p < 0.05$) to the variance of the community data. The stress value of the NMDS plot is 0.13. Each dot represents the mean value of relative OTU abundances from two technical replicates. Sample depth is denoted as meters below the surface for terrestrial samples and meters below the sea floor for submarine samples.

| | 16S Bacteria | 16S/DNA | TCC | Temp | Salinity | Depth [mbs/mbsf] | $Ba^{2+}$ | $Ca^{2+}$ | $K^+$ | $Mg^{2+}$ | $Na^+$ | $Cl^-$ | $SO_4^{2-}$ | $Br^-$ | $\delta^{18}O$ | $\delta D$ | pH | TC | TOC | Grav. Water Content |
|---|---|---|---|---|---|---|---|---|---|---|---|---|---|---|---|---|---|---|---|---|
| DNA | 0.87 | 0.47 | 0.68 | -0.37 | -0.35 | 0.30 | -0.08 | -0.09 | -0.39 | -0.32 | -0.39 | -0.43 | -0.14 | -0.41 | -0.37 | -0.33 | -0.44 | 0.40 | 0.34 | 0.47 |
| 16S Bacteria | | 0.79 | 0.61 | -0.24 | -0.48 | 0.51 | 0.03 | -0.20 | -0.49 | -0.46 | -0.57 | -0.56 | -0.26 | -0.54 | -0.38 | -0.33 | -0.52 | 0.44 | 0.39 | 0.47 |
| 16S / DNA | | | 0.36 | -0.12 | -0.63 | 0.47 | 0.04 | -0.47 | -0.55 | -0.60 | -0.71 | -0.67 | -0.40 | -0.66 | -0.16 | -0.11 | -0.54 | 0.21 | 0.19 | 0.26 |
| TCC | | | | -0.64 | -0.44 | 0.26 | -0.38 | -0.23 | -0.42 | -0.37 | -0.50 | -0.52 | -0.09 | -0.50 | -0.37 | -0.37 | -0.28 | 0.06 | 0.14 | 0.16 |

**Table 1: Spearman correlations of DNA concentration, 16S rRNA gene copy numbers (g$^{-1}$ sediment) and total cell counts (TCC) with environmental and geochemical parameters.** Presented is the correlation coefficient $r_s$. Significant negative correlations are highlighted in red and significant positive correlations are highlighted in green. Colour intensity represents the significance levels, from dark to light colour: $p < 0.001$; $p < 0.01$; $p < 0.05$. P-values and more data can be found in Table S10.