# Peer review of "Microbial community composition and abundance after millennia of submarine permafrost warming"

_Biogeosciences, 2019_

## Referee Comment (RC1) · Anonymous Referee #1 · 11 May 2019

Mitzscherling et al present an interesting study on the microbial communities living in permafrost underneath seawater on the continental shelf. The sampling campaign is quite impressive and extensive, four different drill cores were analyzed in an on shore to offshore transect. Multiple depths from the previously deposited permafrost layer were taken from each core for a comparison based on qPCR, cell counts, 16S rRNA gene sequencing, and various geochemical proxies. The size and integrated nature of the data make this certainly at a level that should be published in Biogeosciences. My main comments are related to methods details that need to be added, additional suggestions for figures (putting the qPCR and cell count data for every sample on an x-y plot), and more discussion of the very interesting findings. I think that the authors are sitting on a

one-of-a-kind dataset, and the discussion as it is reads a bit general and does not do the data justice. With a bit more detailed discussion, the authors could possibly make some interesting links of the microbial groups to past paleo-ecological conditions. For example, I think they should discuss more about what their data mean for the assembly (or lack thereof) of deep biosphere communities in subseafloor sediments. I think that after a minor revision the paper should be suitable for publication.

Specific comments

Abstract:

line 29: Not clear what you mean by "...DOC content was least" (please also define DOC on first use).

line 32: Stable isotopes of what? Carbon?

line 34: Any Fungi?

Methods:

page 3, lines 18 - 26: Did you perform any contamination controls for the drilling? Or is this not necessary because no drill fluid was used? Please explain in the text.

page 4, lines 18-21: What depths do these sections correspond to?

page 4, lines 24-25: Where did you sub sample the core? In a laminar flow clean hood or just on the bench? Are qPCR values high enough that major contamination issues are not a concern? This seems to be the case since you are around 10^7. If yes, please state in the text.

page 5, lines 18-20: Do these primers also target archaea? please specify in the text.

page 6, lines 5-6: These are rather unconventional primers for microbiome studies. Why did you chose them over say the Earth Microbiome primers (515F/806R)? Please explain whether your primers also targeted archaea or not.

page 6, lines 9-10: Please explain in more detail your pipeline for picking 16S OTUs. OTUs are not clustered using the SILVA database, just taxonomy assigned. More information is needed here on how you processed the data, quality control, clustering methods, etc.

page 9, line 6: Which isotopes? 18O? The community is not formed by the isotopes, but probably reflects something else that the isotopes are a proxy for. What is the proxy the water isotopes are showing? Paleo temperature? I thought this was supposed to be related to temperature? But below you say diversity is not related to temperature. Kind of confusing.

page 9, line 14: Again, please explain what the 18O and delta D isotopes are proxies for.

page 9, line 16-17: Since the samples all derive from different depths (at least this is what I gather looking at figure 2), how do you know that temperature is explaining the difference. Do all the depths have the same temperature? Or do the depths from each site have their own unique temperature range? This needs a lot of clarification in the text.

page 9, lines 25-30: Maybe I missed this, but what is physical state of the subsurface samples you acquired via drilling. Is it hard ice , or more slushy? e.g., has it thawed since being overlain with seawater? And, the samples from the terrestrial site are presumably colder, and harder, than those overlain with warmer seawater? If you have any photographs of the cores themselves showing these differences i suggest including them as a figure in the main text. This has important implications for preservation of organics as discussed here.

page 10, lines 9-13: Here, and throughout the text, when you refer to qPCR data can you please actually state in the text what the number of gene copies is? Instead of saying "Low gene copies...". I don't know what you mean by the word "low".

page 10, lines 24-25: Please also cite some of the recent studies showing an influence of paleoclimate on microbial abundance and diversity in sediments (doi.org/10.1038/s41598-017-05590-9 and doi.org/10.1093/femsec/fiy029). This supports your findings here, which is very interesting.

page 12, lines 3-5: Do your qPCR and cell count data correlate ? What is the strength of the correlation? Please add this to the results and show this on an X-Y plot (cell counts vs. qPCR values for all samples) as a new main figure in the text. In the X-Y plot you can give the different points different colors showing which core they derive from. This will be highly interesting and informative !!!

Figures:

Figure 4: Please add the x-y plot I have suggested above. All qPCR and cell count data per sample (it looks like you have a lot!) should be plotted against one another on an x y plot. All individual datapoint should be shown so that readers can see the spread in the data. This will be a major benefit to the paper, improving its strength.

Figure 5: This is a great figure and shows some remarkable patterns. For example, the Atribacteria seem restricted to C3. This was only superficially discussed in the text. What is known about Atribacteria and their ecology, that can explain this? They apparently dominate the entire community in C3. You could discuss this, in the context of the recent review on their metabolism and ecology in the subseafloor (doi: 10.1038/s41579-018-0046-8).

---

## Referee Comment (RC2) · Anonymous Referee #2 · 12 Jul 2019

general comments

The manuscript of Mitzcherling et al. describes a field survey in the arctic, which tested the hypothesis that the effect of permafrost warming can be examined already before the thawing starts. Intriguingly, the experimental design was to use frozen sediment cores of diverging base temperatures ranging between -12° and -1.4°C, but from the approximately same age. This is a very clever setup, however, the implementation of this was limited by using only four sites (maybe because of the costs and logistics of such an expedition), which also limits the statistics and the final conclusions. The authors tried to compensate the limited sampling by taking 6-10 replicates (which could

be statistically interpreted as pseudoreplicates) per site from the targeted frozen period, but in the end could only see a moderate, and even negative effect of temperature on the microbial parameters. Furthermore, other factors such as depth and potential differences in the palaoenvironmental origin obscured the temperature signal, which the authors discussed, accordingly. In general, the study is cleverly designed and provides new research concepts for studying permafrost changes over long time scales. Although the results leave room for discussions due to the limited sampling sites, this work is an interesting study, and a good basis for future studies within the same setting.

specific comments

Page 4 – Age measurements: The authors estimated the age of the core profiles, but it wasn't included in the statistics (or at least I couldn't find it); I would assume that age could explain part of the variation in the microbial community composition.

Page 5 – DNA extraction: Could the authors maybe in the supplement provide a gel picture of the extracted DNA? I am asking this, since the fragmentation of the DNA can also be seen as an indicator for the presence of non-cellular ancient DNA (aDNA). In addition, could the authors please specifiy which size fraction was extracted from gel?

Page 6 – HTS: Please keep in mind that 35 PCR cycles is an unusual high cycle number for an amplicon based microbiome analysis, which will probably cause larger shifts in relative abundance values of microbial groups (this may become relevant when you try to implement some of the RC1 comments).

Page 7 – Multivariate Statistics: To me the authors used a suboptimal set of statistical methods for analysing the microbial community composition. While Mantel tests are good for testing the correlation of two matrices (e.g. the community matrix with the environmental matrix or a subset of parameters (e.g. a matrix of depth, temp, stable isotopes), it is rather uncommon to use it for testing single parameters. (As a side mark, the Mantel test can be performed rank-based or parametric based, this should be specified). Current alternatives to Mantel tests are PERMANOVA variants that are

suitable for continuous variables, or for a priori hypothesis such as the temperature hypothesis distance based redundancy analysis followed by an ANOVA test. However, since the authors start with an exploratory analysis, one of the most frequent used methods is to fit the variables into the ordination by regression/correlation. CCA may not be the best option in this case, but a PCoA or an NMDS would be the preferred method.

Page 7 – statistics: Isn't the Dunn's test the PostHoc test for non-parametric tests such as Kruskal-Wallice? For an ANOVA, I would have expected a Tukey-HSD. Please doublecheck.

Page 7 – General statistics: The authors will need to think about corrections for multiple comparisons. In particular in Table 1 or for the Mantel tests presented in the supplemental, which will both require p-value corrections (e.g. using Bonferroni). If these corrections are not done, this has to be stated explicitly.

Page 8 – Line 16: Please state the correlation values between DNA, copy numbers, and cell counts. This may be important to interpret Table 1.

Page 9 – curiosity comment: The authors took a vertical profile of each core, but this is not really implemented in the study. Out of curiosity: Do the points in e.g. the CCA or the PCA also structure according to the vertical profile? If so, this could be an interesting aspect that may also explain some of the variance observed, caused by differences in ages and/or the paleaoenvironment.

Page 9 – Discussion: The discussion is rather comprehensive and understandable. I think, I can agree with the arguments of the authors. Please include a brief discussion on the limited sampling design of only 4 sites. With such a high variation between the cores, it may require > 30 cores to really answer the hypothesis.

Figures:

Optional comment: Since DOC became important for the discussion of the cell counts

(and equivalents), maybe it could be worthwhile to include a figure on this in the main text. Please discuss this among yourselves.

technical corrections

Page 6 – please specifiy which Illumina MiSeq chemistry was used (2x 250 or 2x300 nt?)

Page 6 – brackets are falsely set in line 13 for Llobet-Brossa et al. 1998

Page 7 – line 19: Please indicate the PSU of the seawater in this area

Figures:

Figure 4: Please indicate the number of measurements for each box (n= . . .)

[Figure]

---

## Author Comment (AC1) · 15 Aug 2019

**Response to Reviewer Comments on Manuscript bg-2019-144:**
**"Microbial community composition and abundance after millennia of submarine permafrost warming"**

*Comments of Reviewer #1:*

Mitzscherling et al present an interesting study on the microbial communities living in permafrost underneath seawater on the continental shelf. The sampling campaign is quite impressive and extensive, four different drill cores were analyzed in an on shore to offshore transect. Multiple depths from the previously deposited permafrost layer were taken from each core for a comparison based on qPCR, cell counts, 16S rRNA gene sequencing, and various geochemical proxies. The size and integrated nature of the data make this certainly at a level that should be published in Biogeosciences. My main comments are related to methods details that need to be added, additional suggestions for figures (putting the qPCR and cell count data for every sample on an x-y plot), and more discussion of the very interesting findings. I think that the authors are sitting on a one-of-a-kind dataset, and the discussion as it is reads a bit general and does not do the data justice. With a bit more detailed discussion, the authors could possibly make some interesting links of the microbial groups to past paleo-ecological conditions. For example, I think they should discuss more about what their data mean for the assembly (or lack thereof) of deep biosphere communities in subseafloor sediments. I think that after a minor revision the paper should be suitable for publication.

We are glad about the positive feedback of referee #1 and the constructive comments on our manuscript. The comments and suggestions contribute to an improvement of the manuscript and we are happy to implement them in the text.

Regarding the general comments, especially the wish for discussing "more about what [our] data mean for the assembly (or lack thereof) of deep biosphere communities in subseafloor sediments", we added the following paragraph to our discussion (page 11 line 11).

Our data suggest that the bacterial community in submarine permafrost sediments has experienced a weak selection after deposition and mostly reflects the paleo-environmental and climatic conditions. Thereby this study joins a number of other studies reporting on microbial groups that are referred to as "the paleome". Those studies found correlations between the microbial diversity and past depositional conditions (Lyra et al., 2013; Orsi et al., 2017; Vuillemin et al., 2016). Marine communities were found in terrestrial settings or soil communities in (sub)seafloor sediments (Ciobanu et al., 2012; Inagaki et al., 2015; Inagaki and Nealson, 2006). Like those, our study implies that the bacterial communities in permafrost soils under the seafloor underwent a weak selection pressure after burial either through dormancy or very low generation times under freezing conditions.

**Abstract:**

line 29: Not clear what you mean by "...DOC content was least" (please also define DOC on first use).

Century-scale permafrost warming is accompanied with decreasing microbial abundance i.e. total cell counts and 16S rRNA gene copies. This is expressed through decreasing abundance from the onshore permafrost core C1, over the offshore cores C4 to C3. In contrast, in the outermost core C2, which experienced warming not only for centuries but for ~2500 years, the abundance increased again. Looking at the dissolved organic carbon content in each of the cores draws a different picture. The lowest DOC values of all cores were found in C2, which experienced warming for millennia. Highest DOC contents were found in core C4, which had lowest 16S rRNA gene copies and TCC that were comparable to core C2.

To clarify this, we rephrased the sentence as follows:

"On time-scales of centuries, permafrost warming coincided with an overall decreasing microbial abundance **whereas** millennia after warming microbial abundance was similar to cold onshore permafrost. **In addition, the dissolved organic carbon** content **of all cores** was **lowest in submarine permafrost after millennia-scale warming**."

**line 32: Stable isotopes of what? Carbon?**

Meant are the stable isotopes of water, i.e. of hydrogen and oxygen – $\delta^{18}O$ and $\delta D$. For more clarity we rephrased the sentence as follows:

"Bacterial community composition correlated only weakly with temperature but strongly with porewater stable isotope**s $\delta^{18}O$ and $\delta D$,** and **with** depth. "

**line 34: Any Fungi?**

Only the bacterial community composition was investigated in this study. The community composition refers to the bacterial community mentioned in the sentence before. We rephrased the sentence as follows.

"Bacterial community composition correlated only weakly with temperature but strongly with porewater stable isotope signatures and depth. **The bacterial** community showed substantial spatial variation and an overall dominance of Actinobacteria, Chloroflexi, Firmicutes, Gemmatimonadetes and Proteobacteria which are amongst the microbial taxa that were also found to be active in other frozen permafrost environments."

**Methods:**

**page 3, lines 18 - 26: Did you perform any contamination controls for the drilling? Or is this not necessary because no drill fluid was used? Please explain in the text.**

Drilling was performed by rotary drilling without using drilling mud as described in page 3 line 22.

Drilling was performed with a hydraulic rotary-pressure **system (Drilling Technologies Factory, St. Petersburg, Russia, Model URB-2A-2) and without the use of any drilling fluid.** All samples were frozen immediately after recovery and were kept at -22 °C until further processing.

Thus, a contamination control of drill fluid was not necessary. Nevertheless, a possible contamination on the rim of the core caused by the drilling equipment should be circumvented by taking subsamples from the center of the core. In order to explain this we added the following information to the text on page 4 lines 24-25:

"**In order t**o **prevent** contamination **caused by the drilling equipment** we took the subsamples from the center of the core"

**page 4, lines 18-21: What depths do these sections correspond to?**

Detailed information about the depth location of each sample can be found in table S4. The approximate depth of each sample and of Unit II within each core is visualized in figure 2a. Numerical data on the depth of Unit II in each core and its extent can be found in table S1 describing each borehole location.

In order to give more details on the sample depths we will add the following information:

"For molecular analyses we took 6 replicate samples from each of the cores C1 (C1-1 – C1-6), C4 (C4-1 – C4-6) and C3 (C3-1 – C3-6) and 8 replicates from core C2 (C2-1, C2-2, C2-4, C2-5, C2-7, C2-8, C2-9, C2-10) (Fig. 2a). Those replicates were located at different depths within Unit II (Table S4). **Samples from C1 were located around 27 to 44 meters below surface, while samples from C4 were taken between 13 and 30 meters below the seafloor, samples from C3 between 9 and 25 m bsf, and samples from C2 between 40 and 58 m bsf**."

**page 4, lines 24-25: Where did you sub sample the core? In a laminar flow clean hood or just on the bench? Are qPCR values high enough that major contamination issues are not a concern? This seems to be the case since you are around 10ˆ7. If yes, please state in the text.**

The core was subsampled in a climate chamber under freezing conditions and by using sterile tools. Thus, contamination can be excluded and are not of concern for downstream analyses like qPCR.

We added these information to the text (page 4 line 25):

Subsampling was performed in a climate chamber under freezing conditions by using sterile tools. Thus, a contamination of the samples can be excluded.

**page 5, lines 18-20: Do these primers also target archaea? please specify in the text.**

According to SILVA those primers do not target archaea when binding without mismatches. Allowing one mismatch only 0.6% of archaeal sequences are targeted.

"Quantitative PCR was performed using the CFX Connect™ Real-Time PCR Detection System (Bio-Rad Laboratories, Inc.) and the primers S-D-Bact-0341-b-S-17 and S-D-Bact-0517-a-A-18 **targeting** the bacterial 16S rRNA gene (Table S5)."

**page 6, lines 5-6: These are rather unconventional primers for microbiome studies. Why did you chose them over say the Earth Microbiome primers (515F/806R)? Please explain whether your primers also targeted archaea or not.**

Microbial biomass in extreme environments like permafrost is known to be low. This can be seen for example in the submarine permafrost core C3. The project from which this manuscript is part of aimed at studying the bacterial and archaeal communities (Mitzscherling et al., 2017; Winkel et al., 2018). However, the archaeal community in permafrost accounts for only a very low percentage of the total microbial community (Hoj et al., 2008; Kobabe et al., 2004). A combined primer pair like the universal 515F/806R may discriminate the amplification of archaeal sequences in presence of an overwhelming amount of bacterial sequences. Hence, we decided to amplify both communities separately. Thus, the primer pair used in this study to investigate the bacterial community was not aimed to target archaeal sequences.
According to SILVA this primer pair covers 0.1% of archaea when assuming no mismatches of the primers. With one mismatch those primers still cover only 1.6% of archaeal sequences.
We agree that the information about the primers target is missing and added the information to the text:
"**The sequencing** primers **that were used in this study only target bacteria and** comprised different combinations of barcodes (Table S6). PCR amplification was carried out with a T100™ Thermal Cycler (Bio-Rad Laboratories, CA, USA). The PCR mixtures (25 µl) contained 1.25 U of OptiTaq DNA Polymerase (Roboklon), 10x concentrate buffer C (Roboklon), 0.5 µM of the sequencing primers S-D-Bact-0341-b-S-17 and S-D-Bact-0785-a-A-21 (Table S5), dNTP 5 mix (0.2 mM each), additional 0.5 mM of $MgCl_2$ (Roboklon), PCR-grade water, and 2.5 µl of template DNA."

page 6, lines 9-10: Please explain in more detail your pipeline for picking 16S OTUs. OTUs are not clustered using the SILVA database, just taxonomy assigned. More information is needed here on how you processed the data, quality control, clustering methods, etc.

Thank you for this remark. Adding these information to the text instead of referencing them helps the reader to get the important information about the sequence analysis and bioinformatical tools right away. Due to the wish of reviewer 2 to specifiy which Illumina MiSeq chemistry was used, we furthermore added detailed information about the sequencing preparation and procedure as well. Thus, we rearranged the methods section as follows:

**2.6 Total cell counts**
…
**2.7 High throughput Illumina16S rRNA gene sequencing**
Sequencing of each sample was performed in two technical replicates. Primers comprised different combinations of barcodes (Table S6). PCR amplification was carried out with a T100™ Thermal Cycler (Bio-Rad Laboratories, CA, USA). The PCR mixtures (25 µl) contained 1.25 U of OptiTaq DNA

Polymerase (Roboklon), 10x concentrate buffer C (Roboklon), 0.5 µM of the sequencing primers S-D-Bact-0341-b-S-17 and S-D-Bact-0785-a-A-21 (Table S5), dNTP mix (0.2 mM each), additional 0.5 mM of MgCl$_2$ (Roboklon), PCR-grade water, and 2.5 µl of template DNA. **PCR conditions comprised an initial denaturation at 95°C for 5 min, followed by 35 cycles of denaturation (95°C for 30 s), annealing (56°C for 30 s) and elongation (72°C for 1 min), and a final extension step of 72°C for 10 min. The PCR products were purified from agarose gel with the HiYieldPCR Clean-Up and Gel-Extraction Kit (Südlabor, Gauting, Germany) and were quantified with the QBIT2 system (Invitrogen, HS-Quant DNA). They were mixed in equimolar amounts and sequenced from both directions (GATC Biotech, Konstanz) based on the Illumina MiSeq technology. The library was prepared with the MiSeq Reagent Kit V3 for 2× 300 bp paired-end reads. The 15% PhiX control v3 library was used for better performance due to different sequencing length.**

2.8 Sequence analysis and bioinformatics

The data analysis of raw bacterial sequences started with the quality control of the sequencing library by the tool FastQC (Quality Control tool for High Throughput Sequence Data http://www.bioinformatics.babraham.ac.uk/projects/fastqc/ by S. Andrews). The tool CutAdapt [Martin, 2011] was used to demultiplex the sequence reads according to their barcodes and to subsequently remove the barcodes. Forward and reverse sequenced fragments with overlapping sequence regions were merged using PEAR [J. Zhang et al., 2014], and the nucleotide sequence orientation was standardized. Low-quality sequences were filtered and trimmed by Trimmomatic [Bolger et al., 2014], and chimeras were removed by Chimera. Slayer. Finally, the QIIME pipeline was used to cluster sequences into operational taxonomic units (OTUs) and to taxonomically assign them employing the SILVA database (release 123) with a cutoff value of 97% [Caporaso et al., 2010].

2.9 Statistics

…

page 9, line 6: Which isotopes? 18O? The community is not formed by the isotopes, but probably reflects something else that the isotopes are a proxy for. What is the proxy the water isotopes are showing? Paleo temperature? I thought this was supposed to be related to temperature? But below you say diversity is not related to temperature. Kind of confusing.

Those are valid questions and show that we did not explain this well enough. The interpretation of these parameters, however, is part of the discussion. The relationship to stable isotopes tells us about paleo-temperature at deposition; temperature of the sediment tells us about the evolution of the microbial environment today, subsequent to deposition. In this paragraph the term "temperature" refers to the temperature of the permafrost sediments at the time of drilling. We hope that this will become clear when adding the following information to the text.

"However, the community formation was stronger influenced by pore water stable isotopes **δ$^{18}$O and δD** (p = 0.0001, R = 0.40) and sample depth (p = 0.0001, R = 0.36), than by **permafrost** temperature."

page 9, line 14: Again, please explain what the 18O and delta D isotopes are proxies for.

We rephrased the sentence as follows:

Variance of samples from the bottom left to the top right was explained by rising **permafrost** temperature, while variance of samples from the top left to the bottom right are likely explained by decreasing values of the stable water isotopes $\delta^{18}O$ and $\delta D$, **a proxy for paleo-temperature and – climate.**

**page 9, line 16-17: Since the samples all derive from different depths (at least this is what I gather looking at figure 2), how do you know that temperature is explaining the difference. Do all the depths have the same temperature? Or do the depths from each site have their own unique temperature range? This needs a lot of clarification in the text.**

Yes, each site has its own unique temperature range. The temperature differences within each core are smaller than across the cores. A description of the temperature range of each core can be found in the Results section (page 7, line 12-15) as well as in the supplementary table S2. Also figure 2b shows the vertical temperature profile of each core.

In order to make this clear we rephrased the sentence …

The variance between C1, C4 and C2 samples are explained by **the** temperature differences **of the permafrost across the cores (Fig. 2b).**

… and rearranged the "Study Site and Drilling" section on page 3 as follows:

The study area (~73°60'N, 117°18'E) is situated in the western part of the Laptev Sea, on the East Siberian Arctic Shelf (Fig. 1). Mean annual bottom water temperatures in the Laptev Sea range between -1.8 °C to -1 °C (Wegner et al., 2005) leading to sediment temperatures of -1.0 °C and -2.0 °C within the largest part of the shelf (Romanovskii et al., 2004). We investigated four cores (C1-C4, Fig. 2a) that were retrieved along an onshore-offshore transect in the coastal region of Cape Mamontov Klyk in 2005 (Overduin, 2007; Rachold et al., 2007**). Cores were named after the order of drilling and we kept this order (C1, C4, C3, C2) for better comparability with previous studies (Koch et al., 2009; Mitzscherling et al., 2017; Overduin et al., 2008; Winkel et al., 2018). From onshore to offshore all cores were characterized by an increase in water depth, in depth to the ice-bonded permafrost table (Fig. 2a, Table S1) and in permafrost temperature (Table S2) (Overduin, 2007; Rachold et al., 2007). The transect was characterized by a temperature gradient that covered an increment of more than 10 °C compared to the onshore permafrost. Thereby, each core displayed its own unique temperature range (Fig. 2b).**

Assuming a constant mean annual coastal erosion rate of 4.5 m yr$^{-1}$ (Grigoriev, 2008) the drill site located furthest offshore (C2, 11.5 km off the coast) was inundated approximately 2500 years ago (Rachold et al., 2007). Accordingly, the drill sites C3 and C4, located 3 km and 1 km off the coast, were inundated around 660 and 220 years ago, respectively. More recent analysis based on remote sensing shows that 40-year coastal erosion rates for the same stretch of coastline between 1965 and 2007 were slower (about 2.9 m yr$^{-1}$) (Günther et al., 2013), which would translate into even longer inundation periods. However, in the present study we refer to Grigoriev (2008), which are based on direct observations of coastal erosion at the C1 coring site. **Drilling was performed with a hydraulic rotary-pressure system (Drilling Technologies Factory, St. Petersburg, Russia, Model URB-2A-2) and without the use of any drilling fluid. All samples were frozen immediately after recovery and were kept at -22 °C until further processing.** Temperature measurements at all sites were done using thermistors and infra-red sensors (Junker et al., 2008).

**page 9, lines 25-30: Maybe I missed this, but what is physical state of the subsurface samples you acquired via drilling. Is it hard ice , or more slushy? e.g., has it thawed since being overlain with seawater? And, the samples from the terrestrial site are presumably colder, and harder, than those overlain with warmer seawater? If you have any photographs of the cores themselves showing these differences i suggest including them as a figure in the main text. This has important implications for preservation of organics as discussed here.**

As described in the section 2.2 samples were selected from Unit II. This "lithostratigraphic Unit II was identified in all cores (Fig. 2a) and was entirely located within the ice-bonded permafrost." Unit II has not been thawed since inundation; it was only warmed from about -12 °C (C1) to around -1 °C (C3 and C2). The colder onshore permafrost has less liquid water than the warmer submarine permafrost as a result.  Terrestrial permafrost is therefore probably somewhat "harder", although this has not been measured and will depend on other factors as well, especially sediment characteristics.
Winterfeld et al., (2011) states that the cryostructure of Unit II in all cores was characterized mainly by pore ice cementing the sediment (gravimetric ice content 20–50 wt%) and also showed typical features of terrestrial permafrost such as segregated ice lenses, ice veins bordering on wood fragments, and composite sand–ice wedges.  We added this information to the paragraph "2.2 Sample selection".
Photographs of Unit II from the terrestrial core C1 and the submarine core C2 were published by Winterfeld et al., (2011). We refrained from publishing them again but referred to them it in our text.

"Each of the four drill cores exhibited different sedimentological units. Lithostratigraphic Unit II was identified in all cores (Fig. 2a) and was entirely located within the ice-bonded permafrost. **Irrespective of the permafrost temperature Unit II sediments of all cores were cemented mainly by pore ice but were also characterized by terrestrial permafrost features like ice lenses, ice veins and ice-wedges. Photographs of (Winterfeld et al., 2011) show similar ice and sediment structures of the terrestrial core C1 and the outermost submarine core C2.** Depth location of Unit II within each core can be found in Table S1."

**page 10, lines 9-13: Here, and throughout the text, when you refer to qPCR data can you please actually state in the text what the number of gene copies is? Instead of saying "Low gene copies...". I don't know what you mean by the word "low".**

We added the order of magnitude when cell counts or gene copy numbers were mentioned in the discussion as follows (page 10 line 30 – page 11 line 11):

The cores C3 and C4 had significantly lower TCC and bacterial gene copy numbers **($10^6$ cells and $10^5$ gene copies)** than the onshore core C1 and the C2 **($10^7$ cells and $10^6$ gene copies)** core furthest offshore. Thus, microbial activity and substrate utilization were likely low in C3 and C4. A negative influence of permafrost warming on microbial abundance is further challenged through some indication for microbial proliferation in core C2, which had experienced longest warming of all cores. In detail, TCC in C2 were higher than in the other submarine cores while DOC values were lower in C2, significantly different from C4 and C1 (Table S13). Permafrost warming for more than two millennia may have enabled microbial communities to adapt to the new temperature regime and sediment properties as suggested before (Mitzscherling et al., 2017). A direct effect of permafrost warming on microbial abundance was not evident; the effect of changing pore-water salinity is more plausible than that of permafrost warming. Rising salinity correlates significantly both with TCC and bacterial gene copy numbers. Also, bacterial 16S rRNA gene copy numbers were lowest in core C4

**(10$^5$ gene copies),** where pore-water salinities were elevated (electrical conductivity values >2000 µS cm$^{-1}$, Table S3). Low gene copy numbers **(10$^5$ gene copies)** may result from osmotic stress that limits microbial growth (Galinski, 1995; Rousk et al., 2011) and decreases microbial abundance in sediments (Jiang et al., 2007; Rath and Rousk, 2015; Rietz and Haynes, 2003; Wen et al., 2018).

**page 10, lines 24-25: Please also cite some of the recent studies showing an influence of paleoclimate on microbial abundance and diversity in sediments (doi.org/10.1038/s41598-017-05590-9 and doi.org/10.1093/femsec/fiy029). This supports your findings here, which is very interesting.**

Thank you for the great suggestion of literature, which perfectly substantiates of our findings. We added the following information at a later stage of the discussion on page 11 in line 4.

We suggest that microbial community composition like microbial abundance reflects the paleoclimate and sedimentation history and not a direct effect of permafrost warming. In detail, we observed a weak correlation between community composition with **permafrost** temperature and a strong correlation with stable water isotope values and depth, i.e. age. **This suggestion is supported by similar findings in sea sediments as well as in lacustrine sediments. Microbial taxa of Arabian Sea sediments reflected past depositional conditions and exhibited paleo-environmental selection (Orsi 2017), while the microbial population in sediments of Laguna Potrok Aike in Argentina changed in response to both past environmental conditions and geochemical changes during burial (Vuillemin, 2018)**.

**page 12, lines 3-5: Do your qPCR and cell count data correlate ? What is the strength of the correlation? Please add this to the results and show this on an X-Y plot (cell counts vs. qPCR values for all samples) as a new main figure in the text. In the X-Y plot you can give the different points different colors showing which core they derive from. This will be highly interesting and informative !!!**

As also reviewer 2 asked for the correlation values between DNA, copy numbers, and cell counts, we added detailed information about p-values and correlation coefficients to table S10 and to the table 1 showing the results of the rank-based Spearman's correlation.

| | 16S Bacteria | 16S/DNA | TCC | Temp | Salinity | Depth [mbsl] | Depth [mbs/ mbsf] | Ba²⁺ | Ca²⁺ | K⁺ | Mg²⁺ | Na⁺ | Si$_{aq}$ | Cl⁻ | SO₄²⁻ | Br⁻ | NO₃⁻ | δ18O | δD | pH | TC | TN | TS | TOC | Clay | Silt | Sand | Grav. Water Content |
|---|---|---|---|---|---|---|---|---|---|---|---|---|---|---|---|---|---|---|---|---|---|---|---|---|---|---|---|---|
| | | | | | | | | | | | | | p-value | | | | | | | | | | | | | | | |
| DNA | >0.001 | >0.001 | >0.001 | 0.030 | 0.039 | 0.813 | 0.076 | 0.658 | 0.604 | 0.020 | 0.061 | 0.021 | 0.872 | 0.011 | 0.410 | 0.015 | 0.593 | 0.027 | 0.055 | 0.008 | 0.017 | 0.329 | 0.175 | 0.045 | 0.307 | 0.111 | 0.130 | 0.006 |
| 16S Bacteria | | >0.001 | >0.001 | 0.173 | 0.003 | 0.164 | 0.002 | 0.860 | 0.248 | 0.003 | 0.005 | >0.001 | 0.475 | >0.001 | 0.128 | 0.001 | 0.587 | 0.023 | 0.054 | 0.002 | 0.009 | 0.175 | 0.056 | 0.021 | 0.821 | 0.886 | 0.926 | 0.007 |
| 16S / DNA | | | 0.03 | 0.503 | >0.001 | 0.216 | 0.004 | 0.799 | 0.005 | 0.001 | >0.001 | >0.001 | 0.268 | >0.001 | 0.016 | >0.001 | 0.425 | 0.369 | 0.528 | 0.001 | 0.218 | 0.171 | 0.135 | 0.284 | 0.055 | 0.102 | 0.084 | 0.153 |
| TCC | | | | >0.001 | 0.008 | 0.465 | 0.138 | 0.024 | 0.193 | 0.012 | 0.029 | 0.002 | 0.262 | 0.001 | 0.593 | 0.002 | 0.890 | 0.028 | 0.027 | 0.097 | 0.749 | 0.759 | 0.233 | 0.429 | 0.184 | 0.572 | 0.524 | 0.369 |
| | | | | | | | | | | | | correlation coefficient r$_s$ | | | | | | | | | | | | | | | | |
| DNA | 0.87 | 0.47 | 0.68 | -0.37 | -0.35 | 0.04 | 0.30 | -0.08 | -0.09 | -0.39 | -0.32 | -0.39 | -0.03 | -0.43 | -0.14 | -0.41 | 0.09 | -0.37 | -0.33 | -0.44 | 0.40 | 0.17 | -0.23 | 0.34 | 0.18 | 0.27 | -0.26 | 0.47 |
| 16S Bacteria | | 0.79 | 0.61 | -0.24 | -0.48 | 0.24 | 0.51 | 0.03 | -0.20 | -0.49 | -0.46 | -0.57 | -0.12 | -0.56 | -0.26 | -0.54 | 0.09 | -0.38 | -0.33 | -0.52 | 0.44 | 0.23 | -0.33 | 0.39 | -0.04 | 0.03 | -0.02 | 0.47 |
| 16S / DNA | | | 0.36 | -0.12 | -0.63 | 0.21 | 0.47 | 0.04 | -0.47 | -0.55 | -0.60 | -0.71 | -0.19 | -0.67 | -0.40 | -0.66 | -0.14 | -0.16 | -0.11 | -0.54 | 0.21 | 0.24 | -0.26 | 0.19 | -0.33 | -0.28 | 0.30 | 0.26 |
| TCC | | | | -0.64 | -0.44 | -0.13 | 0.26 | -0.38 | -0.23 | -0.42 | -0.37 | -0.50 | -0.19 | -0.52 | -0.09 | -0.50 | 0.02 | -0.37 | -0.37 | -0.28 | 0.06 | 0.05 | -0.21 | 0.14 | -0.23 | -0.10 | 0.11 | 0.16 |

Table S10

| | 16S Bacteria | 16S/DNA | TCC | Temp | Salinity | Depth [mbs/ mbsf] | Ba²⁺ | Ca²⁺ | K⁺ | Mg²⁺ | Na⁺ | Cl⁻ | SO₄²⁻ | Br⁻ | δ18O | δD | pH | TC | TOC | Grav. Water Content |
|---|---|---|---|---|---|---|---|---|---|---|---|---|---|---|---|---|---|---|---|---|
| **DNA** | 0.87 | 0.47 | 0.68 | -0.37 | -0.35 | 0.30 | -0.08 | -0.09 | -0.39 | -0.32 | -0.39 | -0.43 | -0.14 | -0.41 | -0.37 | -0.33 | -0.44 | 0.40 | 0.34 | 0.47 |
| **16S Bacteria** | | 0.79 | 0.61 | -0.24 | -0.48 | 0.51 | 0.03 | -0.20 | -0.49 | -0.46 | -0.57 | -0.56 | -0.26 | -0.54 | -0.38 | -0.33 | -0.52 | 0.44 | 0.39 | 0.47 |
| **16S / DNA** | | | 0.36 | -0.12 | -0.63 | 0.47 | 0.04 | -0.47 | -0.55 | -0.60 | -0.71 | -0.67 | -0.40 | -0.66 | -0.16 | -0.11 | -0.54 | 0.21 | 0.19 | 0.26 |
| **TCC** | | | | -0.64 | -0.44 | 0.26 | -0.38 | -0.23 | -0.42 | -0.37 | -0.50 | -0.52 | -0.09 | -0.50 | -0.37 | -0.37 | -0.28 | 0.06 | 0.14 | 0.16 |

Table 1

We also added the suggested X-Y plot showing the total cell counts and versus gene copy numbers in different colors which indicate the core they derive from. Both axes are shown in log scale. We will integrate this figure in Figure 4.

[Figure]

**Figures:**

**Figure 4: Please add the x-y plot I have suggested above. All qPCR and cell count data per sample (it looks like you have a lot!) should be plotted against one another on an x y plot. All individual datapoint should be shown so that readers can see the spread in the data. This will be a major benefit to the paper, improving its strength.**

Please see the answer to the comment above.

**Figure 5: This is a great figure and shows some remarkable patterns. For example, the Atribacteria seem restricted to C3. This was only superficially discussed in the text. What is known about Atribacteria and their ecology, that can explain this? They apparently dominate the entire community in C3. You could discuss this, in the context of the recent review on their metabolism and ecology in the subseafloor (doi: 10.1038/s41579-018-0046-8).**

We thank the reviewer for raising those important questions and giving advice to the interesting literature that can perfectly add up information to the description of Atribacteria in the manuscript. We will add the following information to page 11 line 22-24:

[revised manuscript text omitted]

---

## Author Comment (AC2) · 15 Aug 2019

**Response to Reviewer Comments on Manuscript bg-2019-144:**
**"Microbial community composition and abundance after millennia of submarine permafrost warming"**

*Comments of Reviewer #2:*

We would like to thank the reviewer for evaluating our manuscript positively and giving such useful and important advises and comments, especially those which helped to improve our knowledge on statistical analyses and the application of them in this manuscript.

**general comments**

**The manuscript of Mitzcherling et al. describes a field survey in the arctic, which tested the hypothesis that the effect of permafrost warming can be examined already before the thawing starts. Intriguingly, the experimental design was to use frozen sediment cores of diverging base temperatures ranging between -12° and -1.4 °C, but from the approximately same age. This is a very clever setup, however, the implementation of this was limited by using only four sites (maybe because of the costs and logistics of such an expedition), which also limits the statistics and the final conclusions. The authors tried to compensate the limited sampling by taking 6-10 replicates (which could be statistically interpreted as pseudoreplicates) per site from the targeted frozen period, but in the end could only see a moderate, and even negative effect of temperature on the microbial parameters. Furthermore, other factors such as depth and potential differences in the palaoenvironmental origin obscured the temperature signal, which the authors discussed, accordingly. In general, the study is cleverly designed and provides new research concepts for studying permafrost changes over long time scales. Although the results leave room for discussions due to the limited sampling sites, this work is an interesting study, and a good basis for future studies within the same setting.**

**specific comments**

**Page 4 – Age measurements: The authors estimated the age of the core profiles, but it wasn't included in the statistics (or at least I couldn't find it); I would assume that age could explain part of the variation in the microbial community composition.**

We agree with this idea. Including the sediment ages in the statistics is, however, difficult as sediment age was determined in only a small number of sediment horizons which are different from the samples analyzed here. Furthermore, different dating methods were used across cores and the transect, and the age dates of the different methods differed strongly from one another (Winterfeld et al., 2011). However, sediment depth could be seen as an analogue for age and depth explains a large part of the variation in the microbial community.

**Page 5 – DNA extraction: Could the authors maybe in the supplement provide a gel picture of the extracted DNA? I am asking this, since the fragmentation of the DNA can also be seen as an**

**indicator for the presence of non-cellular ancient DNA (aDNA). In addition, could the authors please specifiy which size fraction was extracted from gel?**

We exemplarily provide the following gel picture of two samples from core C2 (CK1210 SDS: 0.05 m bsf, 265 ng/g and CK1247 SDS: 52.7 m bsf, 33.4 ng/g) and add this to the supplement. The two examples show that there is not much fragmentation likely due to constantly freeze-locked conditions. Hence, we used the DNA extracts without gel purification for downstream analyses.

[Figure]

**Page 6 – HTS: Please keep in mind that 35 PCR cycles is an unusual high cycle number for an amplicon based microbiome analysis, which will probably cause larger shifts in relative abundance values of microbial groups (this may become relevant when you try to implement some of the RC1 comments).**

We thank the reviewer for this reminder. In low biomass environments like permafrost it is, however, difficult to amplify a sufficient amount of DNA for sequencing. There are a number of other studies which implemented the same or even a higher number of PCR cycles in their work (Koebsch et al., 2019; Wen et al., 2018; Winkel et al., 2018, 2019).

**Page 7 – Multivariate Statistics: To me the authors used a suboptimal set of statistical methods for analysing the microbial community composition. While Mantel tests are good for testing the correlation of two matrices (e.g. the community matrix with the environmental matrix or a subset of parameters (e.g. a matrix of depth, temp, stable isotopes), it is rather uncommon to use it for testing single parameters. (As a side mark, the Mantel test can be performed rank-based or parametric based, this should be specified). Current alternatives to Mantel tests are PERMANOVA variants that are suitable for continuous variables, or for a priori hypothesis such as the temperature hypothesis distance based redundancy analysis followed by an ANOVA test. However, since the authors start with an exploratory analysis, one of the most frequent used methods is to fit the variables into the ordination by regression/correlation. CCA may not be the best option in this case, but a PCoA or an NMDS would be the preferred method.**

We thank the reviewer for giving these important advices. We changed the analyses as suggested and changed the text and figures accordingly:

The description of the statistical analysis in the methods part was changed as follows (page 6 line 29):

Variation in $OTU_{0.03}$ composition, 16S rRNA gene and total cell abundance between samples and among drill sites, as well as correlations of the abundance and $OTU_{0.03}$ composition with environmental parameters were assessed using the Past 3.14 software (Hammer et al., 2001) and **R, especially the vegan and MASS packages.** Principal component analyses (PCA) based on Euclidean distance were used to assess variation in environmental variables across the different sediment units and within Unit II. Prior to analysis, all environmental data were standardized by subtracting the mean and dividing by standard deviation. To assess the correlations of bacterial and microbial abundance with environmental parameters the rank-based Spearman correlation was calculated. **The Bray-Curtis dissimilarity was used to assess the beta diversity of the microbial communities in a non-metric multidimensional scaling (NMDS) plot. Environmental factors that might influence its composition were determined by an environmental fit into the ordination. The significance of the variance introduced by the identified environmental factors was tested using a permutational approach as implemented in the adonis function of the vegan package. Factors were tested for auto-correlation as implemented in the corrplot package. A linear model of the remaining factors was subject to a redundancy analysis which was tested for significance using the analysis of variance (ANOVA).**

The produced NMDS plot replaced the CCA in Figure 6.

[Figure]

Figure caption: Non-metric multidimensional scaling (NMDS) plot of $OTU_{0.03}$ data from Unit II in dependence on environmental parameters. Shown are environmental factors that contribute significantly ($p < 0.05$) to the variance of the community data. The stress value of the NMDS plot is 0.13.

Environmental factors that might influence the microbial community composition were tested by Permutational MANOVA. This table replaced the Mantel tests in table S11:

| | Dim1 | Dim2 | $r^2$ | p-value |
|---|---|---|---|---|
| Depth [mbs/msbf] | -0.53174 | -0.84691 | 0.3322 | 0.006 |
| Temperature | 0.13632 | 0.99067 | 0.2487 | 0.015 |
| Ba | 0.94807 | 0.31805 | 0.1859 | 0.031 |
| Si | 0.90304 | -0.42956 | 0.1541 | 0.056 |
| Ca | 0.50032 | 0.86584 | 0.01 | 0.835 |
| K | 0.81761 | 0.57578 | 0.0612 | 0.341 |
| Mg | 0.80879 | 0.58809 | 0.0684 | 0.297 |
| Na | 0.99177 | 0.12804 | 0.0813 | 0.241 |
| Nitrate | -0.8121 | 0.58351 | 0.028 | 0.637 |
| Chloride | 0.98966 | 0.14344 | 0.0527 | 0.391 |
| Sulfate | -0.28689 | 0.95796 | 0.1014 | 0.161 |
| Bromide | 0.92727 | 0.37439 | 0.0629 | 0.326 |
| Salinity | 0.99532 | 0.0966 | 0.0459 | 0.443 |
| δ18O | 0.99329 | -0.11569 | 0.3753 | 0.001 |
| δD | 0.9843 | -0.17648 | 0.3914 | 0.001 |
| pH | -0.42785 | 0.90385 | 0.6412 | 0.001 |
| TC | 0.41379 | -0.91037 | 0.1053 | 0.149 |
| TN | -0.38942 | -0.92106 | 0.0268 | 0.640 |

| | | | | |
|---|---|---|---|---|
| **TS** | **0.03653** | **0.99933** | **0.2694** | **0.004** |
| TOC | 0.40692 | -0.91346 | 0.0974 | 0.170 |
| Clay | 0.47503 | 0.87997 | 0.1123 | 0.132 |
| Silt | 0.76336 | 0.64597 | 0.0532 | 0.405 |
| Sand | -0.70792 | -0.70629 | 0.063 | 0.330 |
| Conductivity | 0.98987 | 0.14199 | 0.0419 | 0.478 |

As the orientation and the distribution of the OTU data changed slightly, we adjusted the description of the results as follows (page 9 line 3-17):

Grouping patterns of the bacterial community based on the $OTU_{0.03}$ composition of the samples and the Bray-Curtis dissimilarity were visualized **using a non-metric multidimensional scaling (NMDS, Fig. 5)**. The **NMDS** showed a clustering of samples according to their borehole location for C2 and C3, while communities of C1 and C4 were more scattered. **We fitted environmental gradients with the NMDS ordination in order to test for correlation between the bacterial community compositions at each drill site with environmental parameters ($p < 0.05$). Samples located at the bottom left of the plot originated from a greater depth (C1 and C2) than samples to the top right (C3 and C4). Variance of samples from the bottom to the top was explained by rising pH, temperature and total sulphur content, while variance of samples from the left to the right side are likely explained by increasing values of the stable water isotopes $\delta^{18}O$ and $\delta D$, and $Ba^{2+}$. The bacterial community of C3 was most distinct and clustered furthest from communities of all other sites, and was linked with stable water isotopes, $Ba^{2+}$ and sample depth. The variance between C1, C4 and C2 samples are explained by temperature differences. A subsequent permutational analysis of variance showed that depth, temperature, pH, TS, $\delta D$, $\delta^{18}O$, and $Ba^{2+}$ contribute to the variance in the microbial community composition (Table S11), whereof $\delta^{18}O$ and $\delta D$ show a high auto-correlation. A redundancy analysis showed that the explanatory variables depth, temperature, pH and $\delta^{18}O$ significantly explain parts of the variance in the microbial composition ($p = 0.001$).**

Finally, we added some discussion on the pH as one of major factors shaping the bacterial community in submarine permafrost samples (page 11 line 7).

The strongest correlation of the bacterial community composition was, however, found with pH. Soil pH is a major factor controlling the bacterial diversity, richness and community composition on a continental scale (Fierer and Jackson, 2006; Lauber et al., 2009; Rousk et al., 2010). On a global scale pH is also one of the major controls of archaeal communities (Wen et al., 2017). Fierer and Jackson (2006) showed that the richness and diversity of bacterial communities differed between ecosystem types, which could be explained by pH. This substantiates our suggestion that Unit II and the bacterial community therein was formed under different paleo-climatic conditions and varying landscape types during the last glacial cycle.

**Page 7 – statistics: Isn't the Dunn's test the PostHoc test for non-parametric tests such as Kruskal-Wallice? For an ANOVA, I would have expected a Tukey-HSD. Please doublecheck.**

We thank the reviewer for pointing to this mistake. It is absolutely right, that the post-hoc test of ANOVA should be the Tukey's test. We changed this accordingly in the supplementary:

**Table S1:** Analysis of variance (ANOVA) of DOC concentrations between all four cores and Tukey's pairwise post-hoc test with p-values according to Copenhaver-Holland above and the Tukey's Q below the diagonal.

| | Sum of squares | df | Mean square | F | p (same) |
|---|---|---|---|---|---|
| Between groups: | 24714.2 | 3 | 8238.06 | 4.814 | 0.003712 |
| Within groups: | 155731 | 91 | 1711.34 | | Permutation p (n=99999) |
| Total: | 180446 | 94 | | | 0.02357 |

| | C1 | C4 | C3 | C2 |
|---|---|---|---|---|
| C1 | | 0.066 | 0.996 | 0.052 |
| C4 | 3,540 | | 0.299 | **0.002** |
| C3 | 0.310 | 2.490 | | 0.739 |
| C2 | 3.676 | 5.209 | 1.441 | |

… and in the text page 10 line 27 :

In detail, TCC in C2 were higher than in the other submarine cores while DOC values were lower in C2, **significantly different from C4 and C1** (Table S13).

**Page 7 – General statistics: The authors will need to think about corrections for multiple comparisons. In particular in Table 1 or for the Mantel tests presented in the supplemental, which will both require p-value corrections (e.g. using Bonferroni). If these corrections are not done, this has to be stated explicitly.**

We thank the reviewer for this remark. As the Mantel tests were replaced by a PerMANOVA we performed a p-value correction only on table 1 according to Holm. The results roughly reflected the original results but with less significant p-values. The correlation of abundance measures to salinity became restricted to 16S rRNA gene copies normalized to DNA $g^{-1}$. We weakened corresponding statements about the significance in the text and rephrased the following sentences.

Abstract (page 1 line 29):
Based on correlation analysis TCC unlike bacterial gene abundance showed a significant rank-based negative correlation with increasing temperature while bacterial gene copy numbers showed a strong negative correlation with salinity

Discussion (page 10 line 8)
Besides permafrost warming changing pore-water salinity had an effect on the microbial abundance. Rising permafrost temperature strongly correlates with TCC whereas salinity correlates strongest with bacterial gene copy numbers (Table 1). Bacterial 16S rRNA gene copy numbers were lowest in core C4 ($10^5$ gene copies), where pore-water salinities were elevated (electrical conductivity values >2000 µS cm$^{-1}$, Table S3).

We furthermore indicated in the caption of table S10 that a p-value correction was not performed here.

Values in bold are significant (< 0.05) when omitting p-value corrections.

**Page 8 – Line 16: Please state the correlation values between DNA, copy numbers, and cell counts. This may be important to interpret Table 1.**

As also reviewer 1 asked for the correlation between the abundance measures we added detailed information about p-values and correlation coefficients to table S10 and the corresponding information to table 1 showing the results of the rank-based Spearman's correlation.

| | 16S Bacteria | 16S/DNA | TCC | Temp | Salinity | Depth [mbs/mbsf] | Ba²⁺ | Ca²⁺ | K⁺ | Mg²⁺ | Na⁺ | Cl⁻ | SO₄²⁻ | Br⁻ | δ¹⁸O | δD | pH | TC | TOC | Grav. Water Content |
|---|---|---|---|---|---|---|---|---|---|---|---|---|---|---|---|---|---|---|---|---|
| DNA | 0.87 | 0.47 | 0.68 | -0.37 | -0.35 | 0.30 | -0.08 | -0.09 | -0.39 | -0.32 | -0.39 | -0.43 | -0.14 | -0.41 | -0.37 | -0.33 | -0.44 | 0.40 | 0.34 | 0.47 |
| 16S Bacteria | | 0.79 | 0.61 | -0.24 | -0.48 | 0.51 | 0.03 | -0.20 | -0.49 | -0.46 | -0.57 | -0.56 | -0.26 | -0.54 | -0.38 | -0.33 | -0.52 | 0.44 | 0.39 | 0.47 |
| 16S / DNA | | | 0.36 | -0.12 | -0.63 | 0.47 | 0.04 | -0.47 | -0.55 | -0.60 | -0.71 | -0.67 | -0.40 | -0.66 | -0.16 | -0.11 | -0.54 | 0.21 | 0.19 | 0.26 |
| TCC | | | | -0.64 | -0.44 | 0.26 | -0.38 | -0.23 | -0.42 | -0.37 | -0.50 | -0.52 | -0.09 | -0.50 | -0.37 | -0.37 | -0.28 | 0.06 | 0.14 | 0.16 |

**Page 9 – curiosity comment: The authors took a vertical profile of each core, but this is not really implemented in the study. Out of curiosity: Do the points in e.g. the CCA or the PCA also structure according to the vertical profile? If so, this could be an interesting aspect that may also explain some of the variance observed, caused by differences in ages and/or the paleoenvironment.**

For the CCA (now NMDS) this is unfortunately not the case. The sample points do not show any gradient according to depth location. In the PCA it is hardly visible if there is any structure according to the vertical profile, especially in the cluster of samples from Unit II.

**Page 9 – Discussion: The discussion is rather comprehensive and understandable. I think, I can agree with the arguments of the authors. Please include a brief discussion on the limited sampling design of only 4 sites. With such a high variation between the cores, it may require > 30 cores to really answer the hypothesis.**

We added a short discussion on the limited sampling size that followed the discussion on the correlation of bacterial community composition and pH.

However, the limited number of environmental samples and the inference of other correlating environmental factors might decrease the statistical powers to see a more significant effect of temperature on the microbial community.

**Figures:**
Optional comment: Since DOC became important for the discussion of the cell counts (and equivalents), maybe it could be worthwhile to include a figure on this in the main text. Please discuss this among yourselves.

We are happy about the reviewers suggestion but decided to leave the figure in the supplemental as we already have a sufficient number of figures in the main text. The supplemental is open and easily accessible for everybody who is interested in more details.

**technical corrections**

**Page 6 – please specifiy which Illumina MiSeq chemistry was used (2x 250 or 2x300nt?)**

Due to the wish of reviewer 1 for more detailed information on the sequence analysis and bioinformatical tools we decided to also add details on the PCR conditions, library preparation and sequencing instead of referencing them. We changed the paragraph as follows:

**2.7 High throughput Illumina16S rRNA gene sequencing**
Sequencing of each sample was performed in two technical replicates. Primers comprised different combinations of barcodes (Table S6). PCR amplification was carried out with a T100™ Thermal Cycler (Bio-Rad Laboratories, CA, USA). The PCR mixtures (25 µl) contained 1.25 U of OptiTaq DNA Polymerase (Roboklon), 10x concentrate buffer C (Roboklon), 0.5 µM of the sequencing primers S-D-Bact-0341-b-S-17 and S-D-Bact-0785-a-A-21 (Table S5), dNTP mix (0.2 mM each), additional 0.5 mM of $MgCl_2$ (Roboklon), PCR-grade water, and 2.5 µl of template DNA. **PCR conditions comprised an initial denaturation at 95°C for 5 min, followed by 35 cycles of denaturation (95°C for 30 s), annealing (56°C for 30 s) and elongation (72°C for 1 min), and a final extension step of 72°C for 10 min. The PCR products were purified from agarose gel with the HiYieldPCR Clean-Up and Gel-Extraction Kit (Südlabor, Gauting, Germany) and were quantified with the QBIT2 system (Invitrogen, HS-Quant DNA). They were mixed in equimolar amounts and sequenced from both directions (GATC Biotech, Konstanz) based on the Illumina MiSeq technology. The library was prepared with the MiSeq Reagent Kit V3 for 2× 300 bp paired-end reads. The 15% PhiX control v3 library was used for better performance due to different sequencing length.**

**Page 6 – brackets are falsely set in line 13 for Llobet-Brossa et al. 1998**

Preparation and quantification of the total cell abundance per g sediment were performed after **Llobet-Brossa et al. (1998)**.

**Page 7 – line 19: Please indicate the PSU of the seawater in this area**

The seawater in the Arctic Ocean is mostly stratified due to the freshwater inflow of large Arctic rivers. Thus, the bottom water salinity is the best to represent the Arctic Ocean waters (Guieu et al., 1996). Bottom-water salinity at the drill sites at Cape Mamontov Klyk were measured in the framework of the drilling campaign and were around 30 PSU. We added the information to the text as follows:
In C4, the drill site located closest to the coast, Unit II had the highest pore water salinity (mean = 5.6 PSU) ranging from 0.9 to 17.6 PSU (Table S2), which spans freshwater to mesohaline water but is much below seawater salinities. **In comparison, bottom-water salinities at the drill sites ranged between 29.2 and 32.2 PSU (Overduin et al., 2008)**.

**Figures:**

**Figure 4: Please indicate the number of measurements for each box (n= : : :)**

We are happy to provide the number of samples for each box. The number of technical replicates measured for each sample is provided in the figure caption.

---

## Author Response (AR1)

**Response to Editor Comment on Manuscript bg-2019-144:**
**"Microbial community composition and abundance after millennia of submarine permafrost warming"**

Dear Denise Akob,

We are pleased about the acceptance of our paper for publication with minor revisions. We are happy
10   to provide the revised manuscript and supplementary material according to our responses to the
referees.

Major changes of the text were made in the following sections:
- Abstract
15   - Materials and Methods (Study Site & Drilling, Sample Selection, High throughput sequencing, we added a paragraph about sequence analysis and bioinformatics)
- Results (bacterial community composition)
- And in the Discussion

20   We replaced Figure 4 and 6 as well as table 1 by new ones.

We added Figure S1 to the supplementary material and replaced the tables S11 and S13.

[revised manuscript text omitted]

**Supplementary Online Material:**

**Contents:**

Figures **S1-S4**: including DNA concentrations and DOC contents and additional information to Figure 2 (PCA)

Tables **S1**-**S14:** including a site description, an overview of selected physicochemical parameters and of the microbial abundances, detailed information about the molecular samples, sequencing primers and barcode sequences, sequencing statistics as well as the results of all statistical tests

[Figure]

**Figure S1:** Quality control of the extracted genomic DNA, exemplarily shown for two samples from core C2 (CK1210 SDS: 0.05 m bsf, 265 ng/g and CK1247 SDS: 52.7 m bsf, 33.4 ng/g). The examples show that there is not much fragmentation likely due to constantly freeze-locked conditions. Hence, we used the DNA extracts without gel purification for downstream analyses.

[Figure]

**Figure S2:** Boxplot of DOC concentrations of Unit II in mg C L-1. Median lines are indicated within the boxes of which the size corresponds to±25% of the data, whereas the whiskers show the minimum and maximum of all data. C1: n=74, C4: n=5, C3: n=3, C2: n =12.

[Figure]

**Figure S3:** DNA concentrations in ng g$^{-1}$ sediment wet weight of the cores C1, C4, C3 and C2. Box plots contain the mean values of all samples, obtained from two technical replicates each. Median lines are indicated within the boxes of which the size corresponds to ± 25% of the data, whereas the whiskers show the minimum and maximum of all data. C1, C4 and C3: n=6, C2: n=17.

[Figure]

**Figure S4:** Loadings plots belonging to Figure 3: PCA of environmental, sedimentological and pore water data from Unit II. Shown are the correlations to a) PC1 and b) PC2 which were used to choose the physicochemical factors that are mainly responsible for the variance between samples in the PCA.

[Figure]

**Figure S5:** Schematic representation of the late Quaternary landscape dynamics in the western Laptev Sea coastal region and formation of the sediments units (modified after Winterfeld et al. (2011)).

**Table S1:** Site description of each borehole location. Mbsf stands for meters below sea floor and dedicates the depth in the submarine cores (C4, C3, C2), while mbs stands for meters below surface and is used for depth indication in the terrestrial core C1.

|  | C1 | C4 | C3 | C2 |
|---|---|---|---|---|
| **Distance to coast [km]** | - 0.1 | 1.0 | 3.0 | 11.5 |
| **Water depth [m]** | - | 2.2 | 4.4 | 6.0 |
| **Frost table depth [mbsf]** | 0.0 | 1.7 | 7.6 | 29.0 |
| **Upper boundary of Unit II [mbs/mbsf]** | 22.0 | 13.3 | 8.6 | 35.0 |
| **Lower boundary of Unit II** | 61.0 | 29.8 | 27.1 | 58.5 |
| **Uppermost sample depth [mbs/mbsf]** | 27.3 | 13.3 | 8.6 | 38.5 |
| **Lowermost sample depth** | 44.3 | 29.8 | 24.9 | 58.4 |

**Table S2:** Minimum, maximum and mean values of environmental factors in Unit II significantly contributing to the bacterial community composition and microbial abundance.

|  | Core | Minimum | Maximum | Mean | Std.Dev. | n |
|---|---|---|---|---|---|---|
| Temperature [°C] | C1 | -12.5 | -12.4 | -12.4 | 0.0 | 8 |
|  | C4 | -7.1 | -5.8 | -6.4 | 0.5 | 4 |
|  | C3 | -1.8 | -1.2 | -1.4 | 0.2 | 4 |
|  | C2 | -1.6 | -1.5 | -1.5 | 0.0 | 4 |
| Salinity [PSU] | C1 | 0.0 | 1.6 | 0.5 | 0.4 | 184 |
|  | C4 | 0.9 | 17.6 | 5.6 | 4.8 | 10 |
|  | C3 | 0.5 | 3.7 | 1.0 | 0.6 | 38 |
|  | C2 | 0.0 | 12.5 | 0.8 | 1.7 | 67 |
| $\delta 18O$ [(‰) vs. SMOW] | C1 | -30.8 | -14.9 | -22.3 | 4.0 | 184 |
|  | C4 | -27.7 | -18.8 | -22.8 | 3.0 | 10 |
|  | C3 | -20.6 | -19.1 | -20.1 | 0.3 | 38 |
|  | C2 | -30.0 | 20.2 | -27.6 | 1.5 | 67 |
| $\delta D$ [(‰) vs. SMOW] | C1 | -241.8 | -115.7 | -177.2 | 32.9 | 184 |
|  | C4 | -219.1 | -144.0 | -178.8 | 25.7 | 10 |
|  | C3 | -162.9 | -149.4 | -158.4 | 2.9 | 38 |
|  | C2 | -232.7 | -156.8 | -213.3 | 11.3 | 67 |

**Table S3:** Geochemical, pore water and environmental data of all samples at each drill site.

See "SuppInfo_Table_3"

**Table S 4:** Sample names of the molecular samples, their depth relative to sea level (meters below sea level, m bsl), relative to surface (meters below surface (m bs) in the terrestrial core and meters below sea floor (m bsf) in the submarine cores), and their corresponding lithology.

| Sample Name | Depth [m bsl] | Depth [m bs/ m bsf] | Lithology |
|---|---|---|---|
| C1-1 | 1.3 | 27.3 | sandy |
| C1-2 | 9.9 | 35.9 | sandy |
| C1-3 | 12.1 | 38.1 | sandy |
| C1-4 | 17.2 | 43.2 | plant/wood detritus |
| C1-5 | 17.4 | 43.4 | plant/wood detritus |
| C1-6 | 18.4 | 44.4 | plant/wood detritus |
| C4-1 | 15.5 | 13.3 | plant/wood detritus |
| C4-2 | 22.0 | 19.8 | small peat inclusions |
| C4-3 | 25.0 | 22.8 | small peat inclusions |
| C4-4 | 28.5 | 26.3 | sandy with quartz gravel |
| C4-5 | 30.0 | 27.8 | sandy with quartz gravel |
| C4-6 | 32.0 | 29.8 | sandy with quartz gravel |
| C3-1 | 13.0 | 8.6 | small peat inclusions |
| C3-2 | 16.0 | 11.6 | small peat inclusions |
| C3-3 | 19.0 | 14.6 | small peat inclusions |
| C3-4 | 21.5 | 17.1 | sandy |
| C3-5 | 24.6 | 20.2 | sandy |
| C3-6 | 29.2 | 24.8 | sandy |
| CK1232 | 44.5 | 38.5 | |
| C2-1 | 46.0 | 40.0 | sandy |
| C2-2 | 48.1 | 42.1 | sandy |
| CK1235 | 49.3 | 43.3 | |
| CK1236 | 50.7 | 44.7 | |
| CK1237 | 51.8 | 45.8 | |
| C2-4 | 52.2 | 46.2 | sandy |
| C2-5 | 54.6 | 48.6 | plant/wood detritus |
| CK 1241 | 54.7 | 48.7 | |
| C2-7 | 55.0 | 49.0 | plant/wood detritus |
| C2-8/1244 | 56.1 | 50.1 | plant/wood detritus |
| CK1245 | 57.8 | 51.8 | |
| CK1246 | 58.2 | 52.2 | |
| CK1247 | 58.7 | 52.7 | |
| CK1248 | 61.6 | 55.6 | |
| C2-9 | 62.9 | 56.9 | plant/wood detritus |
| C2-10 | 64.4 | 58.4 | sandy silt |

**Table S5:** Oligonucleotide primers for Illumina MiSeq sequencing and quantitative PCR.

| Target | Primer Sets | Primer Sequence 5'-3' | Size bp | T (°C) | No. of PCR Cycles | References |
|---|---|---|---|---|---|---|
| **Illumina MiSeq sequencing** | | | | | | |
| Bacterial 16S rRNA | S-D-Bact-0341-b-S-17 | CCT ACG GGA GGC AGC AG | 464 | 55 | 35 | (Muyzer et al., 1993) |
| | S-D-Bact-0785-a-A-21 | GAC TAC HVG GGT ATC TAA TCC | | | | (Herlemann et al., 2011) |
| **Quantitative PCR** | | | | | | |
| Bacterial 16S rRNA | S-D-Bact-0341-b-S-17 | CCT ACG GGA GGC AGC AG | 193 | 55.7 | 40 | (Muyzer et al., 1993) |
| | S-D-Bact-0517-a-A-18 | ATT ACC GCG GCT GCT GG | | | | (Muyzer et al., 1993) |

**Table S6:** Barcode sequences for Illumina MiSeq sequencing.

| Barcode ID Forward Primer | Barcode Sequence | Barcode ID Reverse Primer | Barcode Sequence |
|---|---|---|---|
| Bac-01-For | ACGAGTGCGT | Bac-01-Rev | ACGAGTGCGT |
| Bac-02-For | ACGCTCGACA | Bac-02-Rev | ACGCTCGACA |
| Bac-03-For | AGACGCACT | Bac-04-Rev | AGCACTGTAG |
| Bac-06-For | ATATCGCGAG | Bac-05-Rev | ATCAGACACG |
| Bac-07-For | CGTGTCTCTA | Bac-06-Rev | ATATCGCGAG |
| Bac-08-For | CTCGCGTGT | Bac-07-Rev | CGTGTCTCTA |
| Bac-11-For | TGATACGTCT | Bac-08-Rev | CTCGCGTGTC |
| Bac-13-For | CATAGTAGTG | Bac-11-Rev | TGATACGTCT |
| Bac-15-For | ATACGACGTA | Bac-13-Rev | CATAGTAGTG |
| Bac-16-For | TCACGTACTA | Bac-14-Rev | CGAGAGATAC |
| Bac-17-For | CGTCTAGTA | Bac-17-Rev | CGTCTAGTAC |
| Bac-19-For | TGTACTACT | Bac-18-Rev | TCTACGTAGC |
| Bac-23-For | TACTCTCGTG | Bac-19-Rev | TGTACTACTC |
| Bac-24-For | TAGAGACGAG | Bac-22-Rev | TACGAGTATG |
| Bac-25-For | TCGTCGCTCG | Bac-23-Rev | TACTCTCGTG |
| Bac-26-For | ACATACGCGT | Bac-24-Rev | TAGAGACGAG |
| Bac-27-For | ACGCGAGTAT | Bac-25-Rev | TCGTCGCTCG |
| Bac-28-For | ACTACTATGT | Bac-26-Rev | ACATACGCGT |
| Bac-31-For | AGCGTCGTCT | Bac-28-Rev | ACTACTATGT |
| Bac-33-For | ATAGAGTACT | Bac-30-Rev | AGACTATACT |
| Bac-34-For | CACGCTACGT | Bac-31-Rev | AGCGTCGTCT |
| Bac-35-For | CAGTAGACGT | Bac-33-Rev | ATAGAGTACT |
| Bac-36-For | CGACGTGACT | Bac-34-Rev | CACGCTACGT |
| Bac-38-For | TACACGTGAT | Bac-35-Rev | CAGTAGACGT |
| Bac-39-For | TACAGATCGT | Bac-36-Rev | CGACGTGACT |
| Bac-40-For | TACGCTGTCT | Bac-37-Rev | TACACACACT |
| Bac-41-For | TAGTGTAGAT | Bac-38-Rev | TACACGTGAT |
| Bac-42-For | TCGATCACGT | Bac-39-Rev | TACAGATCGT |
| Bac-44-For | TCTAGCGACT | Bac-40-Rev | TACGCTGTCT |
| Bac-45-For | TCTATACTAT | Bac-41-Rev | TAGTGTAGAT |

| | | | |
|---|---|---|---|
| Bac-49-For | ACGCGATCGA | Bac-44-Rev | TCTAGCGACT |
| Bac-50-For | ACTAGCAGTA | Bac-45-Rev | TCTATACTAT |
| | | Bac-46-Rev | TGACGTATGT |
| | | Bac-49-Rev | ACGCGATCGA |
| SfiA-MW00 | ACACGT | SfiB-MW10 | CAGTCA |
| SfiA-MW01 | ACGTAC | SfiB-MW11 | CATGAC |
| SfiA-MW02 | ACTGCA | SfiB-MW12 | GACTAG |
| SfiA-MW02 | ACTGCA | SfiB-MW13 | GAGATC |
| SfiA-MW03 | AGAGTC | SfiB-MW14 | GATCGA |
| SfiA-MW04 | AGCTGA | SfiB-MW14 | GATCGA |
| SfiA-MW05 | AGTCAG | SfiB-MW15 | GTACAC |
| SfiA-MW06 | ATATCG | SfiB-MW15 | GTACAC |
| SfiA-MW07 | ATCGAT | SfiB-MW16 | GTCACA |
| | | SfiB-MW17 | GTGTGT |
| | | SfiB-MW18 | TCAGAG |
| | | SfiB-MW19 | TCGAGA |

**Table S 7:** Overview of sequencing reads: number of reads after the removal of singletons, number of reads that were removed when the background filter of 0.5% was applied, number of reads representing chloroplast, mitochondrial and archaeal taxa and finally the number of quality reads after the application of all filters. Critical samples with less than 15.000 raw reads are shaded red. Critical samples where the relative abundances within duplicates are comparable are colored light red. The dark red colored sample was not used for the calculation of the mean relative abundance as the relative abundances within duplicates differed.

| | Raw reads | Background Reads (0.5%) | Chloroplast | Mitochondrial Taxa | Archaeal taxa | Quality reads |
|---|---|---|---|---|---|---|
| C1-1a | 74760 | 20214 | 0 | 0 | 0 | 54546 |
| C1-1b | 89992 | 22879 | 0 | 0 | 0 | 67113 |
| C1-2a | 23789 | 15150 | 0 | 0 | 0 | 8639 |
| C1-2b | 25100 | 16330 | 0 | 0 | 0 | 8770 |
| C1-3a | 41727 | 24707 | 0 | 0 | 0 | 17020 |
| C1-3b | 8666 | 5185 | 0 | 0 | 0 | 3481 |
| C1-4a | 31071 | 22620 | 0 | 0 | 0 | 8451 |
| C1-4b | 5142 | 3761 | 0 | 0 | 0 | 1381 |
| C1-5a | 208578 | 100128 | 0 | 0 | 0 | 108450 |
| C1-5b | 147753 | 69180 | 0 | 0 | 0 | 78573 |
| C1-6a | 244866 | 100302 | 0 | 0 | 1790 | 142774 |
| C1-6b | 255535 | 113256 | 0 | 0 | 6331 | 135948 |
| C4-1a | 231425 | 70205 | 0 | 0 | 0 | 161220 |
| C4-1b | 103692 | 30996 | 0 | 0 | 0 | 72696 |
| C4-2a | 312930 | 64767 | 0 | 0 | 0 | 248163 |
| C4-2b | 18603 | 2840 | 0 | 0 | 0 | 15763 |
| C4-3a | 269853 | 99103 | 1765 | 0 | 0 | 168985 |
| C4-3b | 94463 | 33930 | 0 | 0 | 0 | 60533 |
| C4-4a | 170018 | 49851 | 0 | 0 | 1050 | 119117 |
| C4-4b | 180556 | 52147 | 0 | 0 | 0 | 128409 |
| C4-5a | 9823 | 4402 | 0 | 0 | 0 | 5421 |
| C4-5b | 17374 | 7566 | 0 | 0 | 0 | 9808 |
| C4-6a | 56201 | 23871 | 0 | 0 | 0 | 32330 |

| | | | | | | |
|---|---|---|---|---|---|---|
| C4-6b | 20949 | 8951 | 0 | 0 | 0 | 11998 |
| C3-1a | 52885 | 27176 | 0 | 0 | 588 | 25121 |
| C3-1b | 180772 | 94154 | 1165 | 0 | 2402 | 83051 |
| C3-2a | 35528 | 13955 | 0 | 0 | 1511 | 20062 |
| C3-2b | 102122 | 39090 | 652 | 0 | 4780 | 57600 |
| C3-3a | 73935 | 22562 | 0 | 0 | 0 | 51373 |
| C3-3b | 25589 | 7940 | 0 | 0 | 0 | 17649 |
| C3-4a | 53553 | 16543 | 0 | 0 | 1899 | 35111 |
| C3-4b | 22552 | 6552 | 0 | 0 | 398 | 15602 |
| C3-5a | 128366 | 34045 | 1091 | 0 | 0 | 93230 |
| C3-5b | 16643 | 4179 | 152 | 0 | 0 | 12312 |
| C3-6a | 80004 | 20997 | 0 | 0 | 1349 | 57658 |
| C3-6b | 89902 | 21079 | 867 | 0 | 0 | 67956 |
| CK1232-1 | 127284 | 59052 | 0 | 0 | 0 | 68232 |
| CK1232-2 | 161752 | 70043 | 0 | 0 | 0 | 91709 |
| C2-1a | 53571 | 16678 | 0 | 0 | 0 | 36893 |
| C2-1b | 25615 | 7060 | 0 | 0 | 0 | 18555 |
| C2-2a | 55698 | 20602 | 0 | 0 | 0 | 35096 |
| C2-2b | 84301 | 25809 | 0 | 0 | 0 | 58492 |
| CK1235-1 | 206215 | 122462 | 11152 | 0 | 0 | 72601 |
| CK1235-2 | 74429 | 43777 | 4130 | 0 | 0 | 26522 |
| CK1236-1 | 20564 | 11265 | 0 | 0 | 0 | 9299 |
| CK1236-2 | 55725 | 30832 | 0 | 0 | 0 | 24893 |
| CK1237-1 | 102376 | 62617 | 0 | 0 | 0 | 39759 |
| CK1237-2 | 139700 | 86444 | 0 | 0 | 0 | 53256 |
| C2-4a | 136761 | 74460 | 0 | 0 | 0 | 62301 |
| C2-4b | 216318 | 118201 | 0 | 0 | 0 | 98117 |
| C2-5a | 48354 | 27604 | 0 | 0 | 0 | 20750 |
| C2-5b | 92506 | 55950 | 0 | 0 | 0 | 36556 |
| CK1241-1 | 177526 | 115666 | 0 | 0 | 0 | 61860 |
| CK1241-2 | 142667 | 88091 | 0 | 0 | 0 | 54576 |
| C2-7a | 63745 | 36419 | 0 | 0 | 0 | 27326 |
| C2-7b | 159960 | 88818 | 0 | 0 | 0 | 71142 |
| C2-8a | 22420 | 14376 | 0 | 0 | 0 | 8044 |
| C2-8b | 130842 | 85938 | 0 | 0 | 0 | 44904 |
| CK1244-1 | 99934 | 54354 | 0 | 0 | 0 | 45580 |
| CK1244-2 | 15808 | 9077 | 0 | 0 | 0 | 6731 |
| CK1245-1 | 81822 | 42330 | 0 | 0 | 0 | 39492 |
| CK1245-2 | 49130 | 24254 | 0 | 0 | 0 | 24876 |
| CK1246-1 | 52169 | 30142 | 0 | 0 | 0 | 22027 |
| CK1246-2 | 70027 | 43178 | 0 | 0 | 0 | 26849 |
| CK1247-1 | 32592 | 14991 | 0 | 0 | 0 | 17601 |
| CK1247-2 | 21821 | 9398 | 0 | 0 | 0 | 12423 |
| CK1248-1 | 25455 | 16365 | 0 | 0 | 0 | 9090 |
| CK1248-2 | 48980 | 32070 | 0 | 0 | 0 | 16910 |
| C2-9a | 37303 | 24313 | 0 | 0 | 0 | 12990 |
| C2-9b | 43272 | 26410 | 0 | 0 | 0 | 16862 |

| | | | | | |
|---|---|---|---|---|---|
| C2-10a | 1889 | 1288 | 0 | 0 | 0 | 601 |
| C2-10b | 213822 | 155149 | 1128 | 0 | 0 | 57545 |

**Table S 8:** Spearman correlation of DNA concentration, bacterial 16S rRNA gene abundance and total cell counts. P-values are shown above the diagonal and the correlation coefficient $r_s$ below.

| | DNA | 16S rRNA gene copies | TCC |
|---|---|---|---|
| DNA | | **>0.0001** | **>0.0001** |
| 16S rRNA gene copies | 0.87 | | **0.0001** |
| TCC | 0.68 | 0.61 | |

**Table S9:** Minimum. maximum. mean values and standard deviation of microbial and bacterial abundance. n indicates the number of samples.

| | Core | Min | Max | Mean | Std. dev. | n |
|---|---|---|---|---|---|---|
| DNA concentration [ng g$^{-1}$] | C1 | 28.6 | 331.3 | 141.6 | 105.6 | 6 |
| | C4 | 6.2 | 277.5 | 88.5 | 102.6 | 6 |
| | C3 | 5.6 | 51.9 | 19.8 | 17.8 | 6 |
| | C2 | 8.7 | 341.5 | 106.9 | 94.0 | 17 |
| 16S rRNA gene copies [g$^{-1}$ sediment] | C1 | 2.4E+07 | 4.3E+08 | 1.6E+08 | 1.4E+08 | 6 |
| | C4 | 1.7E+06 | 1.6E+08 | 3.6E+07 | 5.8E+07 | 6 |
| | C3 | 4.1E+06 | 6.2E+07 | 1.7E+07 | 2.1E+07 | 6 |
| | C2 | 5.4E+06 | 1.5E+09 | 2.9E+08 | 4.0E+08 | 17 |
| 16S rRNA gene copies [ng$^{-1}$ DNA] | C1 | 7.6E+05 | 1.4E+06 | 1.0E+06 | 2.6E+05 | 6 |
| | C4 | 5.8E+04 | 5.8E+05 | 2.7E+05 | 1.6E+05 | 6 |
| | C3 | 4.9E+05 | 1.2E+06 | 7.6E+05 | 2.2E+05 | 6 |
| | C2 | 2.6E+05 | 1.7E+07 | 2.7E+06 | 4.2E+06 | 17 |
| TCC [g$^{-1}$ sediment] | C1 | 6.8E+06 | 8.2E+07 | 5.0E+07 | 2.8E+07 | 6 |
| | C4 | 1.6E+06 | 4.4E+07 | 1.3E+07 | 1.5E+07 | 6 |
| | C3 | 3.4E+05 | 3.4E+06 | 1.5E+06 | 1.0E+06 | 6 |
| | C2 | 1.4E+06 | 4.9E+07 | 1.5E+07 | 1.3E+07 | 17 |

**Table S10:** Rank-based Spearman correlation of DNA concentration, bacterial 16S rRNA gene abundance and total cell counts with environmental factors and pore water data. Values in bold are significant (< 0.05) when omitting a p-value correction. $R_s$- values highlighted red show a negative correlation, whereas $r_s$-values highlighted green show a positive correlation.

| | 16S Bacteria | 16S/DNA | TCC | Temp | Salinity | Depth [mbsl] | Depth [mbs/mbsf] | $Ba^{2+}$ | $Ca^{2+}$ | $K^+$ | $Mg^{2+}$ | $Na^+$ | $Si_{aq}$ | $Cl^-$ | $SO_4^{2-}$ | $Br^-$ | $NO_3^-$ | $\delta18O$ | $\delta D$ | pH | TC | TN | TS | TOC | Clay | Silt | Sand | Grav. Water Content |
|---|---|---|---|---|---|---|---|---|---|---|---|---|---|---|---|---|---|---|---|---|---|---|---|---|---|---|---|---|
| **p-value** | | | | | | | | | | | | | | | | | | | | | | | | | | | | |
| DNA | **>0.001** | **>0.001** | **>0.001** | **0.030** | **0.039** | 0.813 | 0.076 | 0.658 | 0.604 | **0.020** | 0.061 | **0.021** | 0.872 | **0.011** | 0.410 | **0.015** | 0.593 | **0.027** | 0.055 | **0.008** | **0.017** | 0.329 | 0.175 | **0.045** | 0.307 | 0.111 | 0.130 | **0.006** |
| 16S Bact. | | **>0.001** | **>0.001** | 0.173 | **0.003** | 0.164 | **0.002** | 0.860 | 0.248 | **0.003** | **0.005** | **>0.001** | 0.475 | **>0.001** | 0.128 | **0.001** | 0.587 | **0.023** | 0.054 | **0.002** | **0.009** | 0.175 | 0.056 | **0.021** | 0.821 | 0.886 | 0.926 | **0.007** |
| 16S / DNA | | | **0.03** | 0.503 | **>0.001** | 0.216 | **0.004** | 0.799 | **0.005** | **0.001** | **>0.001** | **>0.001** | 0.268 | **>0.001** | **0.016** | **>0.001** | 0.425 | 0.369 | 0.528 | **0.001** | 0.218 | 0.171 | 0.135 | 0.284 | 0.055 | 0.102 | 0.084 | 0.153 |
| TCC | | | | **>0.001** | **0.008** | 0.465 | 0.138 | **0.024** | 0.193 | **0.012** | **0.029** | **0.002** | 0.262 | **0.001** | 0.593 | **0.002** | 0.890 | **0.028** | **0.027** | 0.097 | 0.749 | 0.759 | 0.233 | 0.429 | 0.184 | 0.572 | 0.524 | 0.369 |
| **correlation coefficient $r_s$** | | | | | | | | | | | | | | | | | | | | | | | | | | | | |
| DNA | 0.87 | 0.47 | 0.68 | -0.37 | -0.35 | 0.04 | 0.30 | -0.08 | -0.09 | -0.39 | -0.32 | -0.39 | -0.03 | -0.43 | -0.14 | -0.41 | 0.09 | -0.37 | -0.33 | -0.44 | 0.40 | 0.17 | -0.23 | 0.34 | 0.18 | 0.27 | -0.26 | 0.47 |
| 16S Bact. | | 0.79 | 0.61 | -0.24 | -0.48 | 0.24 | 0.51 | 0.03 | -0.20 | -0.49 | -0.46 | -0.57 | -0.12 | -0.56 | -0.26 | -0.54 | 0.09 | -0.38 | -0.33 | -0.52 | 0.44 | 0.23 | -0.33 | 0.39 | -0.04 | 0.03 | -0.02 | 0.47 |
| 16S / DNA | | | 0.36 | -0.12 | -0.63 | 0.21 | 0.47 | 0.04 | -0.47 | -0.55 | -0.60 | -0.71 | -0.19 | -0.67 | -0.40 | -0.66 | -0.14 | -0.16 | -0.11 | -0.54 | 0.21 | 0.24 | -0.26 | 0.19 | -0.33 | -0.28 | 0.30 | 0.26 |
| TCC | | | | -0.64 | -0.44 | -0.13 | 0.26 | -0.38 | -0.23 | -0.42 | -0.37 | -0.50 | -0.19 | -0.52 | -0.09 | -0.50 | 0.02 | -0.37 | -0.37 | -0.28 | 0.06 | 0.05 | -0.21 | 0.14 | -0.23 | -0.10 | 0.11 | 0.16 |

**Table S11:** Significance of the variance introduced by environmental factors into the microbial community tested by Permutational MANOVA (PerMANOVA)

| _ | Dim1 | Dim2 | $r^2$ | p-value |
|---|---|---|---|---|
| **Depth [mbs/msbf]** | **-0.53174** | **-0.84691** | **0.3322** | **0.006** |
| **Temperature** | **0.13632** | **0.99067** | **0.2487** | **0.015** |
| **Ba** | **0.94807** | **0.31805** | **0.1859** | **0.031** |
| Si | 0.90304 | -0.42956 | 0.1541 | 0.056 |
| Ca | 0.50032 | 0.86584 | 0.0100 | 0.835 |
| K | 0.81761 | 0.57578 | 0.0612 | 0.341 |
| Mg | 0.80879 | 0.58809 | 0.0684 | 0.297 |
| Na | 0.99177 | 0.12804 | 0.0813 | 0.241 |
| Nitrate | -0.81210 | 0.58351 | 0.0280 | 0.637 |
| Chloride | 0.98966 | 0.14344 | 0.0527 | 0.391 |
| Sulfate | -0.28689 | 0.95796 | 0.1014 | 0.161 |
| Bromide | 0.92727 | 0.37439 | 0.0629 | 0.326 |
| Salinity | 0.99532 | 0.09660 | 0.0459 | 0.443 |
| **δ18O** | **0.99329** | **-0.11569** | **0.3753** | **0.001** |
| **δD** | **0.98430** | **-0.17648** | **0.3914** | **0.001** |
| **pH** | **-0.42785** | **0.90385** | **0.6412** | **0.001** |
| TC | 0.41379 | -0.91037 | 0.1053 | 0.149 |
| TN | -0.38942 | -0.92106 | 0.0268 | 0.640 |
| **TS** | **0.03653** | **0.99933** | **0.2694** | **0.004** |
| TOC | 0.40692 | -0.91346 | 0.0974 | 0.170 |
| Clay | 0.47503 | 0.87997 | 0.1123 | 0.132 |
| Silt | 0.76336 | 0.64597 | 0.0532 | 0.405 |
| Sand | -0.70792 | -0.70629 | 0.0630 | 0.330 |
| Conductivity | 0.98987 | 0.14199 | 0.0419 | 0.478 |

**Table S12:** One-way PerMANOVA of OTU data from each drill site. Summary presents the overall test statistics. Pairwise analysis shows Bonferroni corrected p-values above the diagonal and F-values below.

| Summary | | | Pairwise | | | | |
|---|---|---|---|---|---|---|---|
| | | | | C1 | C4 | C3 | C2 |
| Permutation N: | 9999 | | | | | | |
| Total sum of squares: | 26.49 | | C1 | | 0.0012 | 0.0006 | 0.0006 |
| Within-group sum of squares: | 19.41 | | C4 | 4.014 | | 0.0006 | 0.0006 |
| F: | 8.276 | | C3 | 12.400 | 7.368 | | 0.0006 |
| p (same): | 0.0001 | | C2 | 5.833 | 5.156 | 16.350 | |

**Table S13:** Analysis of variance (ANOVA) of DOC concentrations between all four cores and Tukey's pairwise post-hoc test with p-values adjusted according to Copenhaver-Holland above and the Tukey's Q below the diagonal.

| | Sum of squares | df | Mean square | F | p (same) |
|---|---|---|---|---|---|
| **Between groups:** | 24714.2 | 3 | 8238.06 | 4.814 | 0.003712 |
| **Within groups:** | 155731 | 91 | 1711.34 | **Permutation p (n=99999)** | |
| **Total:** | 180446 | 94 | | | 0.02357 |

| - | C1 | C4 | C3 | C2 |
|---|---|---|---|---|
| C1 | | 0.066 | 0.996 | 0.052 |
| C4 | 3,540 | | 0.299 | **0.002** |
| C3 | 0.310 | 2.490 | | 0.739 |
| C2 | 3.676 | 5.209 | 1.441 | |

**Table S14:** Fossil bioindicators according to Schirrmeister et al. (2008); Winterfeld et al. (2011); Müller et al. (2009) and their stratigraphical and paleoenvironmental interpretation.

| Units | Bioindicator | Stratigraphy | Landscape, facies | Vegetation | Climate |
|---|---|---|---|---|---|
| IId | • Pollen*: Cyperaceae, Poaceae, Artemisia, Salix*
• Spores: *Encalypta, Glomus*
• Green algae: *Botryococcus, Pediastrum*
• Ostracodes
• Plant macro remains: Carex, *Salix* sp., *Saxifraga hirculus*, *Dryas Kobresia myosuroides, Thlaspitea rotundifolii*
• Testacea: hygrophillic (Difflugia), sphagnobiotic (*Heleopera, Nebela, Argynnia* sp.)
• Mammals: *Equus caballus, Mammuthus primigenius* | Middle Weichselian Interstadial | Floodplain, alluvial, boggy, periodically flooded | Grass-sedge tundra | Moderate, humid |
| IIc | No determinable fossil records found | Early Weichselian Stadial | fluvial | | |
| IIb | Pollen: *Larix, Alnus fruticosa, Betula nana*, Ericales | Eemian Interglacial | Thermokarst lake | Shrub tundra | Temperate |
| IIa | No determinable fossil records found | Early Weichselian Stadial | fluvial | | |
| I | • Marine diatoms: *Hyalodiscus* sp*., Paralia sulcata, Porosira glacialis, Thalassiosira* sp*., Thalassiothrix longissima, Centralea* ind.
• Fresh water diatoms: *Naicula radiosa, Eunotia praerupta, Pinnularia gibba, tetracyclus lacustris*
• Sponge spicula
• Pollen: *Larix, Alnus fruticosa, Betula nana*, Ericales | Eemian Interglacial | Thermokarst lagoon | Shrub tundra | temperate |

- Spores: *Sphagnum*

Herlemann, D. P., Labrenz, M., Jürgens, K., Bertilsson, S., Waniek, J. J. and Andersson, A. F.: Transitions in bacterial communities along the 2000 km salinity gradient of the Baltic Sea., ISME J., 5(10), 1571–9, doi:10.1038/ismej.2011.41, 2011.

Müller, S., Bobrov, A. A., Schirrmeister, L., Andreev, A. A. and Tarasov, P. E.: Testate amoebae record from the Laptev Sea coast and its implication for the reconstruction of Late Pleistocene and Holocene environments in the Arctic Siberia, Palaeogeogr. Palaeoclimatol. Palaeoecol., 271(3–4), 301–315, doi:10.1016/J.PALAEO.2008.11.003, 2009.

Muyzer, G., De Waal, E. C. and Uitterlinden, A. G.: Profiling of complex microbial populations by denaturing gradient gel electrophoresis analysis of polymerase chain reaction-amplified genes coding for 16S rRNA., Appl. Environ. Microbiol., 59(3), 695–700, 1993.

Schirrmeister, L., Grosse, G., Kunitsky, V., Magens, D., Meyer, H., Dereviagin, A., Kuznetsova, T., Andreev, A., Babiy, O., Kienast, F., Grigoriev, M., Overduin, P. P. and Preusser, F.: Periglacial landscape evolution and environmental changes of Arctic lowland areas for the last 60000 years (western Laptev Sea coast, Cape Mamontov Klyk), Polar Res., 27(2), 249–272, doi:10.1111/j.1751-8369.2008.00067.x, 2008.

Winterfeld, M., Schirrmeister, L., Grigoriev, M. N., Kunitsky, V. V., Andreev, A., Murray, A. and Overduin, P. P.: Coastal permafrost landscape development since the Late Pleistocene in the western Laptev Sea, Siberia, Boreas, 40(4), 697–713, doi:10.1111/j.1502-3885.2011.00203.x, 2011.

---

## Author Response (AR2)

**Response to Editor Comment on Manuscript bg-2019-144:**

**"Microbial community composition and abundance after millennia of submarine permafrost warming"**

Dear Denise Akob,

We are happy about the acceptance for publication in Biogeosciences. Regarding your question/technical correction for pg. 7, l. 20, we mean of course 15 thousand i.e. 15,000 and changed it accordingly in the text. Thank you for this remark.

Kind regards,

Julia Mitzscherling